# Squeezing More from the Stream : Learning Representation Online for Streaming Reinforcement Learning

Nilaksh [* 1 2 3]   Antoine Clavaud [* 1 2 3]   Mathieu Reymond [1 2]   François Rivest [4 2]   Sarath Chandar [1 2 3 5]

## Abstract

In streaming Reinforcement Learning (RL), transitions are observed and discarded immediately after a single update. While this minimizes resource usage for on-device applications, it makes agents notoriously sample-inefficient, since value-based losses alone struggle to extract meaningful representations from transient data. We propose extending Self-Predictive Representations (SPR) to the streaming pipeline to maximize the utility of every observed frame. However, due to the highly correlated samples induced by the streaming regime, naively applying this auxiliary loss results in training instabilities. Thus, we introduce orthogonal gradient updates relative to the momentum target and resolve gradient conflicts arising from streaming-specific optimizers. Validated across the Atari, MinAtar, and Octax suites, our approach systematically outperforms existing streaming baselines. Latent-space analysis, including t-SNE visualizations and effective-rank measurements, confirms that our method learns significantly richer representations, bridging the performance gap caused by the absence of a replay buffer, while remaining efficient enough to train on just a few CPU cores.

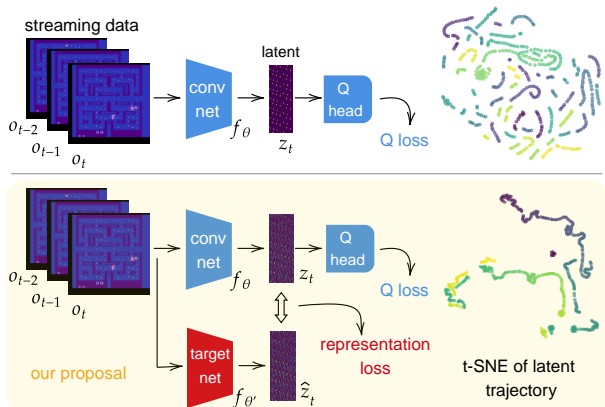

Figure 1. Streaming RL setups so far (TOP) have only relied on the Q Learning loss to learn task specific representations. However we propose using an additional representation learning loss (BOTTOM) to help the encoder learn better representations. The t-SNE visualizations of the latents of a sample trajectory affirm this since our training method produces a smooth latent trajectory, with temporally closer samples also closer in the t-SNE plot as shown by the color gradient. **Takeaway:** Representation learning loss makes a better use of the streaming data than Q learning loss alone, and improves policy performance.

## 1. Introduction

Streaming Reinforcement Learning (RL) is an online learning paradigm where an agent learns from a continuous sequence of experiences that are processed individually and discarded immediately. This setting is essential for compute and memory constrained scenarios such as on-device learning and robotics, as well as for privacy-sensitive ap-

plications, since it does not store huge replay buffers or perform expensive batch updates. Intuitively, the streaming setting can also help the agent adapt quickly to changes in the environment, compared to training on a replay buffer that may contain stale data. Despite these advantages, most modern deep RL algorithms move away from this regime, relying instead on large replay buffers, batch updates, and target networks to stabilize learning.

This shift is not accidental. Streaming RL suffers from poor sample efficiency, as well as unstable updates due non-stationarity. Experience replay mitigates both by reusing samples multiple times and stabilizing updates through gradient averaging over large batch sizes (Mnih et al., 2015; van Hasselt et al., 2016; Hessel et al., 2018; Haarnoja et al., 2018). Recently, new work has shown that deep streaming RL is possible (Vasan et al., 2024) and stable. The Stream-X

*Equal contribution [1]Chandar Research Lab [2]Mila - Quebec AI Institute [3]Polytechnique Montréal [4]Royal Military College of Canada [5]Canada CIFAR AI Chair. Correspondence to: Nilaksh <nilaksh.nilaksh@mila.quebec>.

*Proceedings of the 43rd International Conference on Machine Learning*, Seoul, South Korea. PMLR 306, 2026. Copyright 2026 by the author(s).

family of algorithms achieves stable learning without replay buffers, target networks, or batched updates, by relying on careful architectural and optimization choices, most notably the introduction of a custom optimizer, ObGD (Elsayed et al., 2024). Another prominent algorithm is QRC($\lambda$), which reformulates the learning objective using the Projected Bellman Error derived from $\lambda$-returns (Elelimy et al., 2025). However, both algorithms focus mainly on the stability issues and leave sample efficiency mostly unaddressed, except for the use of eligibility traces. Yet, because in the streaming setting each transition is used only once before being discarded, the already well-known sample inefficiency of RL losses is exacerbated. As a result, streaming RL is severely sample-inefficient: each transition is expensive to obtain, and its informational content is only weakly exploited before being discarded.

In this work, we address this inefficiency by explicitly augmenting streaming RL with a representation learning objective. Our key observation is simple: if data must be discarded after a single update, then it becomes crucial to extract as much information as possible from that update. We therefore supplement the standard RL objective with an auxiliary self-supervised loss that directly trains the encoder to learn more informative representations. As illustrated in Fig. 1, adding this representation learning loss enables the encoder to learn substantially better representations compared to using the RL value-loss alone.

We instantiate this idea using the Self-Predictive Representations (SPR) framework (Schwarzer et al., 2021). SPR is well-suited to the streaming setting because it predicts future latent representations using lightweight prediction heads and a simple latent dynamics model. This improves representation quality without large memory footprints or expensive batch computations, both of which would undermine the core motivation for streaming RL. However, combining distinct SPR and reinforcement learning losses within a non-stationary, highly stochastic setting is not trivial. We address this by orthogonalizing SPR gradients against their own moving average to handle correlated data, and against the primary RL updates to prevent interference.

We use Stream Q($\lambda$) (from the Stream-X family) and QRC($\lambda$) along with a streaming version of DQN as our baselines to test the effectiveness of our SPR-based streaming representation learning framework. We validate our approach on the Atari (Bellemare et al., 2013), MinAtar (Young & Tian, 2019), and Octax (Radji et al., 2026) suites, demonstrating that representation learning is the missing piece for sample efficient streaming RL. Notably, our method remains computationally efficient and can be trained on just a few of CPU cores[1]. Our contributions are as follows:

---

[1] Code Repository: github.com/chandar-lab/stream-rep-rl

1. *SPR for Streaming*: We show that integrating SPR into the streaming setting leads to higher sample efficiency and final returns across diverse environments, improving both QRC($\lambda$) and streaming DQN baselines.

2. *Orthogonal Gradient Updates*: To mitigate the strong temporal correlation of streaming data, we propose an improvement to the base SPR framework. We project the current SPR gradient onto the subspace orthogonal to its gradient history, leaving the RL updates unchanged.

3. *Optimization for ObGD*: We offer an investigation into the gradient conflicts that arise when using the ObGD optimizer with auxiliary losses and provide suitable modifications to ensure stable convergence.

## 2. Related Work

**Representation Learning for RL** Model-free RL is inherently sample inefficient, requiring millions of frames to train on games like Atari (Hessel et al., 2018). It often requires each transition to be stored in a replay buffer so that it can be used to be trained on multiple times. One way to improve sample efficiency is to learn auxiliary tasks (Jaderberg et al., 2016). A common set of auxiliary tasks is to predict some part of the dynamics of the environment (Schwarzer et al., 2021; Tomar et al., 2023; Zhao et al., 2023; Scannell et al., 2024; Fujimoto et al., 2025). Usually, these methods add a secondary objective, such as predicting the next observation(s), next latent state(s), next reward(s), action(s) used (inverse dynamics) or a combination of these, and use the weight sharing in the encoder to boost the policy's performances. Ni et al. (2024) give a good overview of such methods, showing that what matters is to have some form of self predictive representations. Tasks relying on self-supervised contrastive objectives have also been tried (Laskin et al., 2020; Zheng et al., 2024).

Another way to learn representations for RL is to use successor features (Barreto et al., 2017; Ma et al., 2020; Chua et al., 2024). These emerge from the linear decomposition of the Q-value as the product of a (non-linear) representation of the states and a (learned) task vector. The key difference between successor features and self predictive representations is that only the former verify a Bellman optimality equation, meaning they induce a fixed point which can be easier to optimize towards. Nevertheless, a unifying characteristic of most representation learning techniques in RL is the use of large replay buffers to try and overcome non-stationarity and data correlation. Not many works study representation learning in a streaming fashion (Han et al., 2025), and none do for streaming RL, making this an open challenge.

**Continual Learning** Continual learning usually consists in training an agent on multiple tasks sequentially in such

a way that the agent is still good at solving the first tasks it was trained on, while still presenting a good ability to learn new ones. This presents challenges like catastrophic forgetting (McCloskey & Cohen, 1989) and loss of plasticity (Dohare et al., 2024). Common practice to alleviate these is to use replay buffers (Purushwalkam et al., 2022), sparse representations (Meyer et al., 2024) or layer normalization (Ba et al., 2016) among others. Continual learning deals more broadly with learning representations from a stream of data, which is generally seen as a very hard task (Prabhu et al., 2024). Most of these representation learning methods are inspired by the supervised learning (i.i.d.) literature and do not translate well to streaming, non-i.i.d. settings, or RL (Bagus & Gepperth, 2022).

**Deep Streaming RL** Even though early Reinforcement Learning algorithms such as TD($\lambda$), Q($\lambda$) and AC($\lambda$) (Sutton & Barto, 2018) to name a few, were designed for the streaming tabular case, they have since widely been adapted to use non-linear value function approximation while replacing the streaming setting with a replay-buffer-based setting, closer to the i.i.d. setting. Taking Deep RL algorithms back to the streaming context is a much more recent achievement (Elsayed et al., 2024; Vasan et al., 2024; Elelimy et al., 2025; Vincent et al., 2026), and is still an under-studied area of RL, apart from some previous works (Javed et al., 2024). Deep streaming RL suffers from very high non-stationarity which makes the optimization problem much harder to solve. To our knowledge, no other papers have tried to combine representation learning and deep streaming RL.

## 3. Background

**Reinforcement Learning** Reinforcement Learning (RL) models sequential decision-making as a Markov Decision Process (MDP) $(\mathcal{S}, \mathcal{A}, \mathcal{P}, \mathcal{R}, \gamma)$, where $\mathcal{S}$ is the state space, $\mathcal{A}$ the action space, $\mathcal{P}$ the transition dynamics, $\mathcal{R}$ the reward function, and $\gamma$ the discount factor. The agent interacts via a policy $\pi(a|s)$, receiving reward $r_t = \mathcal{R}(s_t, a_t)$ and transitioning to $s_{t+1} \sim \mathcal{P}(\cdot|s_t, a_t)$ at each step $t$.

The episodic return is defined as $G_t = \sum_{k=t}^{T} \gamma^{k-t} r_k$. The state-value function $V_\pi(s) = \mathbb{E}_\pi[G_t|s_t = s]$ evaluates the expected return from state $s$, and the agent's goal is to find an optimal policy $\pi^* = \text{argmax}_\pi\{V_\pi(s_0)\}$. Similarly, the action-value function $Q_\pi(s, a) = \mathbb{E}_\pi[G_t|s_t = s, a_t = a]$ satisfies the Bellman optimality equation: $Q_{\pi^*}(s_t, a_t) = r_t + \gamma \max_a\{Q_{\pi^*}(s_{t+1}, a)\}$.

Deep Q-Learning approximates this function using a neural network $Q_\theta$. Parameters $\theta$ are updated via the semi-gradient of the Mean Squared Bellman Error loss, where SG denotes the *Stop Gradient* operation: $\mathcal{L}(\theta) = (Q_\theta(s_t, a_t) - r_t - \gamma \cdot \text{SG}(\max_a\{Q_\theta(s_{t+1}, a)\}))^2$.

**Streaming RL** The problem statement of RL does not change in the streaming context. Elsayed et al. (2024)'s definition of the streaming setting of Reinforcement Learning mandates that any element of the transition $(s_t, a_t, r_t, s_{t+1})$ must be discarded right before the next transition. Any learning can only happen on the immediately available data. However, we can aggregate any running statistic of the past transitions that we want. For instance, observation and reward normalization are allowed because these operations can be performed online, without storing any transition. This is one of the methods used by Elsayed et al. (2024) in their *Stream Q($\lambda$)* algorithm, of which we give a more detailed explanation in Appendix E. They also use layer normalization (Ba et al., 2016) before each activation function across the entire network, eligibility traces and a custom optimizer, Overshooting-bounded Gradient Descent (ObGD). ObGD performs a one-step approximation of a backtracking line-search algorithm (Armijo, 1966) which finds the highest learning rate that still leads to stable gradient update. QRC($\lambda$) (Elelimy et al., 2025) is another streaming RL algorithm, presented in more details in appendix D, that shares many tricks with Stream Q($\lambda$), such as reward and observation normalization as well as the use of eligibility traces. We diverge a little from Elsayed et al. (2024)'s setting by allowing the use of target networks, as well as to store a small, fixed size buffer, since in a practical streaming setting (like on-device learning), the only real constraint is unavailability of multiple sources of data to train on.

**Eligibility traces** To improve sampling efficiency in a Q-network we can use the $n$-step return. Instead of just considering the next step in the computation of the return, we consider a higher bootstrapping depth, thereby defining $Q_{\pi^*}(s_t, a_t) = \sum_{k=0}^{n-1} \gamma^k r_{t+k} + \gamma^n \max_a\{Q_{\pi^*}(s_{t+n}, a)\}$ as the Bellman optimality equation. Building further on this, the $\lambda$-return combines $n$-step returns for all possible values of $n$, weighted by $\lambda$. This formulation reduces variance compared to a fixed $n$-step return (Daley & Amato, 2019) and naturally prioritizes temporally closer experiences. The $\lambda$-return is implemented using the "backward-view" algorithm which only needs a running statistic of the previously seen states to perform a learning update, thus being stream friendly. This running statistic is called the eligibility trace and is defined as $\mathbf{z}_0 = \mathbf{0}$, $\mathbf{z}_{t+1} = \lambda\gamma\mathbf{z}_t + \nabla_\theta Q_\theta(s_t, a_t)$.

Eligibility traces $\mathbf{z}$ are reset to zero between episodes, and in the case of $\epsilon$-greedy exploration, whenever the action taken is not the greedy one. This leads to the parameter update rule $\theta \leftarrow \theta + \alpha\delta_t\mathbf{z}_t$, where $\delta_t$ is the TD-error (difference between the current and bootstrapped estimate of the Q-value).

**Self-Predictive Representations (SPR)** In a broader setting, self predictive representations are a representation learning technique inspired from model-based RL. They are

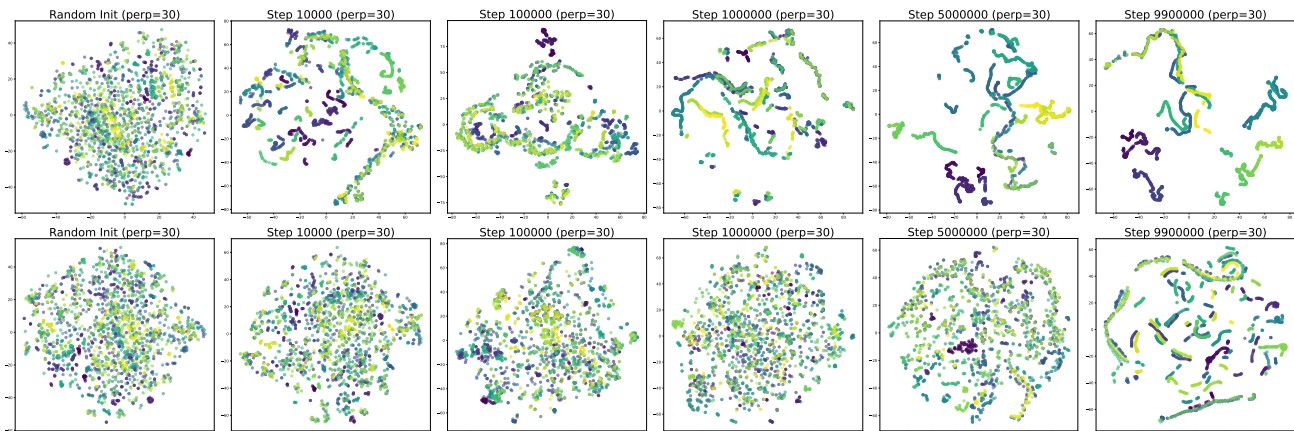

*Figure 2.* We visualize the t-SNE (Maaten & Hinton, 2008) (with a perplexity of 30) of the encoder latents computed on a random 5000 steps trajectory on the Atari Alien environment with the QRC($\lambda$) (Elelimy et al., 2025) agent. They show how the encoder latents evolve over the training, with the first column being the random initialization and the last column being after 40M frames. The color gradient shows the temporal relation between the samples. The SPR latents (TOP) display distinct clusters made up of several continuous segments and evolve much faster than the non-SPR latents (BOTTOM), which either lack these structures or show them late in their training. Figure 9 in Appendix H shows visualizations on other environments and perplexities.

designed to learn rich state ($s_t$) representations $z_t = f_\theta(s_t)$ through a secondary objective, relying on learning a dynamics model $D$ of the environment, usually such that $D(z_t, a_t) = \hat{z}_{t+1}$, where $a_t$ is the action. This prediction can also happen in the observation space rather than the latent space, but this has been show to be less effective as the learned representation must also be able to be used to reconstruct irrelevant details of the image (Zhang et al., 2020; Schwarzer et al., 2021). The encoder is shared between the representation learning and the Q-value parts of the overall network, effectively sharing weights to improve the Q-network's representations. A loss function (e.g. Mean Squared Error (MSE) or cosine similarity) is then applied between the predicted next latent state and the actual next latent state $\mathcal{L}_{SPR} = \text{loss}(D_\psi(f_\theta(s_t), a_t), f_\theta(s_{t+1}))$ so that by learning to predict the dynamics of the environment, the shared encoder learns a good representation of the states. Schwarzer et al. (2021) build upon this idea and sequentially predict the next $K$ latent states and use a cosine similarity loss rather than a reconstruction-oriented MSE loss.

## 4. Method

In order to evaluate the effectiveness of the auxiliary representation learning objective in the streaming setting we consider the Stream Q($\lambda$) agent from (Elsayed et al., 2024), the QRC($\lambda$) agent from (Elelimy et al., 2025), and a modified streaming DQN as our baselines. What we call streaming DQN is a version of Stream Q($\lambda$) which keeps the same stability enhancements, but where we replace the ObGD optimizer by SGD and add a target value-network. This baseline represents a logical first attempt at a streaming agent and also helps study the interaction of ObGD with

SPR. Following Elelimy et al. (2025), we use SGD as our optimizer for QRC($\lambda$) with an $\epsilon$-greedy exploration schedule. More details on QRC($\lambda$) are in Appendix D.

As is done in Stream Q($\lambda$)'s original setting we used parameter-free layer normalization (no scaling or bias parameter is learned) on all the pre-activations and used normalized observations and rewards. We give a more complete overview of our experimental setup, hyperparameter values and implementation choices in Appendix A.

### 4.1. Streaming Self-Predictive Representations

To improve sample efficiency and representation learning in the streaming setting, we augment the agent with a Self-Predictive Representation (SPR) auxiliary task. Figure 1 (BOTTOM) offers a simplified depiction of Schwarzer et al. (2021)'s SPR framework.

Formally, we employ an online encoder $f_\theta$ to map the current observation $o_t$ to a latent state $z_t$, and a momentum-averaged target encoder $f_{\theta'}$ to generate stable targets from future observations. A transition model $D_\psi$ iteratively predicts future latent states $\hat{z}_{t+k}$ conditioned on the sequence of actions $a_{t:t+K-1}$, starting from $\hat{z}_t = z_t$. These latent predictions are mapped through a projection head $P_\phi$ and prediction head $q_\omega$ to produce $\hat{y}_{t+k} = q_\omega(P_\phi(\hat{z}_{t+k})$. Note that these layers are distinct from the Q-Learning head, however $P_\phi$ can optionally share parameters with the head (see Appendix A.7). Finally $\hat{y}_{t+k}$ is compared against the stop-gradient target projections $\tilde{y}_{t+k} = P_{\phi'}(f_{\theta'}(o_{t+k}))$. The total SPR loss is computed as the negative cosine similarity between these predicted and target representations, summed over the horizon $K$:

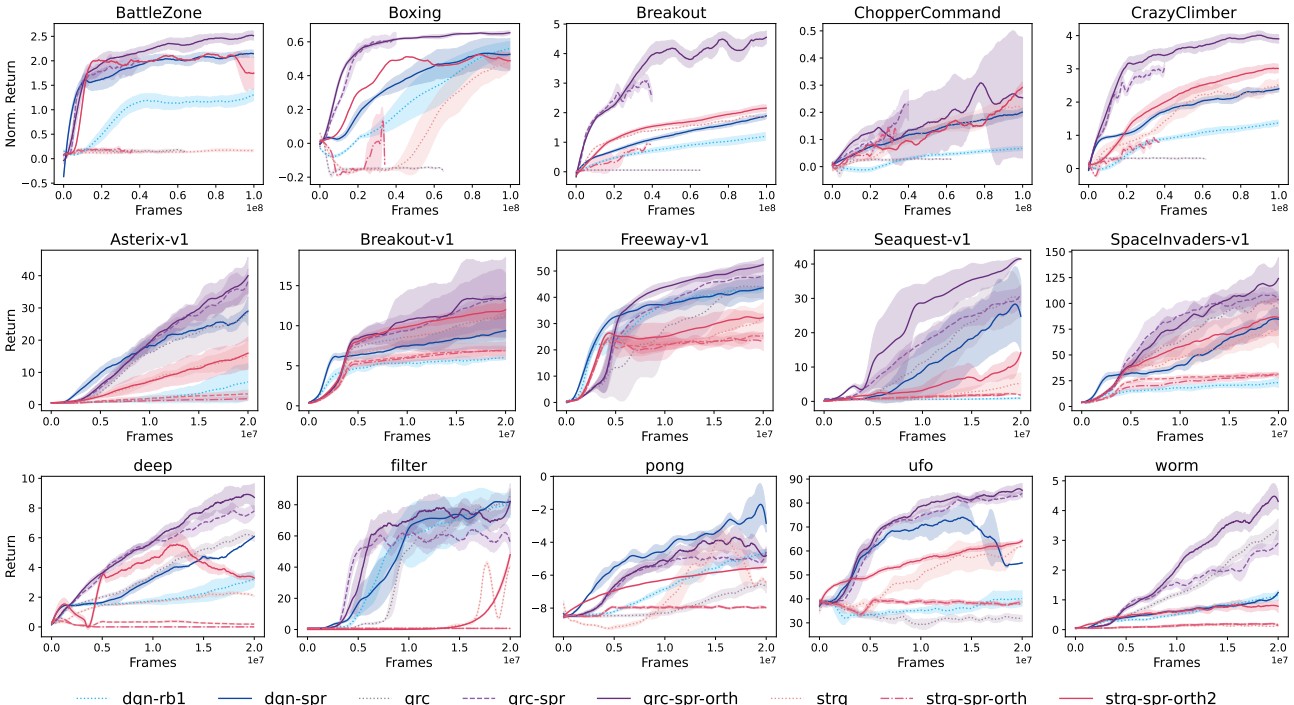

*Figure 3.* The learning curves of the different algorithms on Atari (TOP), MinAtar (MIDDLE) and Octax (BOTTOM). We see that SPR leads to much better sample efficiency for QRC($\lambda$) on all environments, and for streaming DQN on most Atari environments. Stream Q($\lambda$) without modifications turns out to be incompatible with SPR, however adding another orthogonalization step orth$^2$ (Sec. 4.3) improves SPR performance and yields higher returns (Table 1). Please see Sec. 5.1 for a description of the legend names. Figures 7 and 8 in Appendix G show training curves on additional environments.

$$\mathcal{L}^{SPR}(\theta, \psi, \phi, \omega) = -\sum_{k=1}^{K} \left( \frac{\hat{y}_{t+k}}{\|\hat{y}_{t+k}\|_2} \cdot \frac{\text{SG}[\tilde{y}_{t+k}]}{\|\text{SG}[\tilde{y}_{t+k}]\|_2} \right)$$

where the target parameters $\theta'$ and $\phi'$ are updated via an exponential moving average (EMA) of $\theta$ and $\phi$ with the coefficient $\tau$. See Appendix A.7 for details.

We use this framework directly in the streaming setting, except the updates are performed with SGD. We keep image augmentations and $K = 5$ as the next latent state prediction window in SPR as it has been shown to offer improved learning capabilities in harder environments (see Table 5 and Sec. 5, Schwarzer et al. (2021)). Although this entails storing a fixed number $K$ of states, it fits our setting for practical streaming RL, with negligible impact on memory compared to a typical replay buffer.

### 4.2. Decorrelating Updates via Orthogonalization

In the streaming setting, the absence of a replay buffer results in highly temporally correlated transitions that can cause parameters to over-fit on recent trajectories. To address this, we follow Han et al. (2025) by utilizing orthogonal gradients during training to decorrelate updates. This

helps because stochastic optimization methods such as SGD work on the assumption that a random mini-batch of data can approximate an unbiased estimate of the global gradient. This assumption breaks in the streaming setting and thus needs additional regularization. As detailed in Appendix C, this method projects the current gradient $\nabla_\eta \mathcal{L}_t^{SPR}$ onto the plane orthogonal to a momentum-based history of previous updates.

We apply this projection *per-component* by flattening the parameter tree of each module $\eta \in \{\theta, \psi, \phi, \omega\}$ (representing the encoder, transition model, projection head, and prediction head respectively) to compute a single scalar projection coefficient for the entire module. Crucially, this process interacts *independently* with the RL objective; the projection uses a dedicated momentum history $m_t^\eta = \beta_{orth} m_{t-1}^\eta + (1 - \beta_{orth}) \nabla_\eta \mathcal{L}_t^{SPR}$ and is applied only to SPR gradients. These processed gradients are then combined additively with the trace-based RL updates ($\Delta \eta_t^{RL}$), ensuring the decorrelation logic does not alter the accumulated eligibility traces. The final update rule for any parameter $\eta$ becomes: $\eta_{t+1} = \eta_t + \Delta \eta_t^{RL} - \alpha \lambda_{SPR} \tilde{g}_t^\eta$, where $\tilde{g}_t^\eta = \nabla_\eta \mathcal{L}_t^{SPR} - \text{proj}_{m_t^\eta}(\nabla_\eta \mathcal{L}_t^{SPR})$ is the projected gradient, and $\Delta \eta_t^{RL}$ is zero for non-shared parameters $\psi, \phi$, and $\omega$. This orthogonalization acts as a lightweight proxy

for experience replay decorrelation, ensuring that representation learning remains stable and high-rank despite the non-i.i.d. nature of the data stream.

### 4.3. Reconciling ObGD and SGD Gradient Conflict

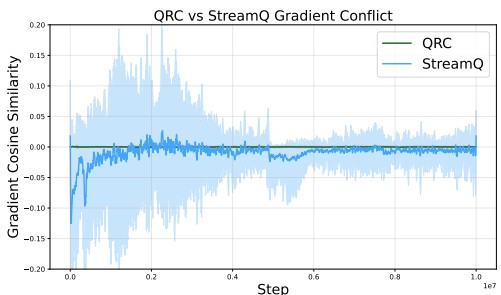

*Figure 4.* The cosine similarity between the gradients back-propagated from the Q network and the SPR networks, averaged over 5 Atari environments. In case of QRC($\lambda$) these come from the SGD optimizer, however Stream Q($\lambda$) uses ObGD for the Q network and SGD for SPR networks. This leads to conflicting gradient updates indicated by the negative cosine similarity, possibly explaining the poor SPR performance with Stream Q($\lambda$). In contrast, for QRC($\lambda$), the gradients are almost orthogonal.

The encoder serves as a shared resource satisfying two objectives: Reinforcement Learning (RL) and representation learning. While QRC($\lambda$) and streaming DQN optimize both using SGD, Stream Q($\lambda$) utilizes ObGD for the RL objective and SGD for SPR. Combining these distinct updates is non-trivial because ObGD and SGD induce different learning-rate dynamics. ObGD adaptively chooses the largest stable step allowed by the current TD error and eligibility trace,

$$\alpha_{\text{ObGD}} \le \frac{1}{\kappa \bar{\bar{\delta}}_t \|\mathbf{z}_t\|_1}, \tag{1}$$

where $\bar{\bar{\delta}}_t = \max(1, |\delta_t|)$ and $\kappa > 1$ is the safety coefficient (see Appx. E for details). This step size can vary substantially across consecutive transitions, whereas SPR is optimized with a fixed SGD learning rate. When ObGD takes a very small step due to a large TD-error or eligibility trace, the fixed-size SPR update can dominate the shared encoder update and move the representation away from the direction for which ObGD's stability bound was calibrated. In contrast, QRC($\lambda$) uses SGD for both losses, so the relative scale of the RL and SPR updates remains much more stable. We first investigated merging the two updates via linear combination:

$$\begin{aligned} \theta_{\text{shared}} &\leftarrow \theta_{\text{shared}} + \mu_{\text{shared}} \alpha_{\text{ObGD}} \delta_t \mathbf{z}_t \\ &- (1 - \mu_{\text{shared}}) \alpha_{\text{SPR}} \lambda_{\text{SPR}} \text{SGD}\big(\nabla \mathcal{L}_{\text{SPR}}(\theta_{\text{shared}})\big) \end{aligned} \tag{2}$$

where $\mu_{\text{shared}}$ (default 0.5) is a mixing coefficient, $\lambda_{SPR}$ is

the SPR loss coefficient (default $\lambda_{SPR} = 2$), and $\alpha_{\text{ObGD}}$, $\alpha_{\text{SPR}}$ are the respective learning rates.

In the streaming setting, these updates often conflict. The cosine similarity between the RL update $\alpha_{\text{ObGD}} \delta_t \mathbf{z}_t$ and the SPR update $-\alpha_{\text{SPR}} \lambda_{\text{SPR}} \nabla_\theta \mathcal{L}_t^{\text{SPR}}$ frequently drops below zero (Fig. 4), indicating the tasks push parameters in opposing directions. This is especially problematic because the ObGD step size in Eq. 1 is computed only from the RL stability criterion. Conversely, for QRC($\lambda$), this gradient conflict is negligible.

To prevent the auxiliary task from interfering with primary policy learning, we employ orthogonal gradient projection (similar to Sec. 4.2). Before updating the encoder, we project the SPR update $u^{SPR}$ onto the subspace orthogonal to the RL update $u^{RL}$ derived from ObGD. The orthogonalized SPR update $\tilde{u}^{SPR}$ is computed as: $\tilde{u}^{SPR} = u^{SPR} - \text{proj}_{u^{RL}}(u^{SPR})$

This ensures representation learning only modifies weights in directions orthogonal to the primary policy update. As shown in our experiments, this reconciliation is important for enabling SPR within the Stream Q($\lambda$) framework. We provide a detailed algorithmic description of this and the previous section in Algorithm 1, Appendix A.10.

### 4.4. Properties of the Orthogonal Updates

Our proposed gradient projection admits a useful local characterization, similar in spirit to gradient-surgery methods (Yu et al., 2020; Liu et al., 2021) for multi-objective optimization. Consider a generic gradient $g_t$ and a reference vector $c_{t-1}$, corresponding either to the SPR gradient history in Sec. 4.2 or to the RL update in Sec. 4.3. The projected gradient $\hat{g}_t = g_t - \text{proj}_{c_{t-1}}(g_t)$ is non-expansive: by the Pythagorean theorem, $\|\hat{g}_t\| \le \|g_t\|$. Moreover,

$$\langle g_t, \hat{g}_t \rangle = \|g_t\|^2 - \frac{\langle g_t, c_{t-1} \rangle^2}{\|c_{t-1}\|^2} \ge 0, \tag{3}$$

by Cauchy–Schwarz, so the projected update never turns a descent direction into an ascent direction. If $g_t$ is parallel to $c_{t-1}$, the projection is zero, as desired when the current gradient is fully redundant with the recent update direction.

The projection introduces a bias $b_t = \hat{g}_t - g_t = -\text{proj}_{c_{t-1}}(g_t)$, but this bias is adaptive:

$$\|b_t\| \le \|g_t\| |\cos(\angle(g_t, c_{t-1}))|. \tag{4}$$

This bias is desired under high temporal correlation (Han et al., 2025). It is largest precisely when consecutive updates are highly aligned, where removing redundant correlated motion is most useful, and becomes small when the data distribution shifts and the new gradient is less aligned with the history.

# 5. Experiments

*Table 1.* The aggregate return metrics with standard deviations, computed over all environments on Atari, MinAtar and Octax. Each experiment was run for five random seeds. We highlight the highest value per metric, however a large variance exits due to the range of returns across individual environments. For a more detailed comparison, please see Tables 9, 10, 11 and 12 in Appendix B. We see that adding SPR (and modifications): streaming DQN (dqn), QRC($\lambda$) (qrc) (Elelimy et al., 2025), and Stream Q($\lambda$) (strq) (Elsayed et al., 2024). Please see Sec. 4.2 (orth) and Sec. 4.3 (orth$^2$) for details on these modifications.

| Algorithm | Mean Return | Median Return | IQM Return |
|---|---|---|---|
| Atari (Human Normalized) - 40M frames | | | |
| qrc | $0.04 \pm 0.28$ | $0.02 \pm 0.28$ | $0.03 \pm 0.03$ |
| qrc+spr | $1.13 \pm 1.19$ | $0.46 \pm 1.18$ | $0.61 \pm 0.39$ |
| qrc+spr+orth | $\mathbf{1.25 \pm 1.55}$ | $\mathbf{0.50 \pm 1.55}$ | $\mathbf{0.62 \pm 0.43}$ |
| strq | $0.48 \pm 0.64$ | $0.15 \pm 0.64$ | $0.23 \pm 0.14$ |
| strq+spr | $0.28 \pm 0.35$ | $0.11 \pm 0.35$ | $0.13 \pm 0.09$ |
| strq+spr+orth | $0.24 \pm 0.29$ | $0.08 \pm 0.29$ | $0.12 \pm 0.09$ |
| strq+spr+orth$^2$ | $0.61 \pm 0.67$ | $0.32 \pm 0.67$ | $0.37 \pm 0.24$ |
| dqn | $0.29 \pm 0.52$ | $0.10 \pm 0.52$ | $0.12 \pm 0.08$ |
| dqn+spr | $0.35 \pm 0.53$ | $0.19 \pm 0.53$ | $0.22 \pm 0.11$ |
| Atari (Human Normalized) - 100M frames | | | |
| qrc | $0.05 \pm 0.28$ | $0.02 \pm 0.28$ | $0.03 \pm 0.03$ |
| qrc+spr+orth | $\mathbf{1.61 \pm 2.05}$ | $\mathbf{0.63 \pm 2.05}$ | $\mathbf{0.80 \pm 0.47}$ |
| strq | $0.79 \pm 0.88$ | $0.40 \pm 0.88$ | $0.44 \pm 0.25$ |
| strq+spr+orth$^2$ | $0.91 \pm 0.93$ | $0.47 \pm 0.93$ | $0.56 \pm 0.34$ |
| dqn | $0.59 \pm 0.76$ | $0.24 \pm 0.76$ | $0.32 \pm 0.23$ |
| dqn+spr | $0.75 \pm 0.79$ | $0.47 \pm 0.79$ | $0.51 \pm 0.28$ |
| MinAtar - 5M frames | | | |
| qrc | $42.50 \pm 30.47$ | $29.01 \pm 30.47$ | $33.94 \pm 7.18$ |
| qrc+spr | $46.66 \pm 32.32$ | $35.75 \pm 32.32$ | $37.58 \pm 7.59$ |
| qrc+spr+orth | $\mathbf{52.70 \pm 36.27}$ | $\mathbf{39.91 \pm 36.27}$ | $\mathbf{43.13 \pm 6.56}$ |
| strq | $27.42 \pm 24.81$ | $15.29 \pm 24.81$ | $19.42 \pm 9.03$ |
| strq+spr | $13.76 \pm 12.09$ | $6.91 \pm 12.09$ | $11.81 \pm 9.66$ |
| strq+spr+orth | $12.86 \pm 11.69$ | $6.87 \pm 11.69$ | $10.84 \pm 9.25$ |
| strq+spr+orth$^2$ | $31.25 \pm 28.81$ | $15.83 \pm 28.81$ | $19.95 \pm 8.78$ |
| dqn | $15.99 \pm 15.46$ | $6.85 \pm 15.46$ | $11.96 \pm 7.80$ |
| dqn+spr | $38.06 \pm 24.98$ | $28.36 \pm 24.98$ | $32.60 \pm 7.80$ |
| Octax - 20M frames | | | |
| qrc | $38.78 \pm 60.56$ | $7.14 \pm 60.56$ | $12.52 \pm 11.48$ |
| qrc+spr | $48.65 \pm 78.10$ | $7.72 \pm 78.10$ | $19.05 \pm 21.98$ |
| qrc+spr+orth | $\mathbf{52.75 \pm 78.96}$ | $\mathbf{8.36 \pm 78.96}$ | $\mathbf{23.16 \pm 28.37}$ |
| strq | $16.43 \pm 23.35$ | $2.82 \pm 23.35$ | $5.98 \pm 7.01$ |
| strq+spr | $16.70 \pm 33.73$ | $0.45 \pm 33.73$ | $1.06 \pm 1.23$ |
| strq+spr+orth | $5.57 \pm 13.34$ | $0.43 \pm 13.34$ | $0.74 \pm 0.83$ |
| strq+spr+orth$^2$ | $26.68 \pm 36.96$ | $3.29 \pm 36.96$ | $13.35 \pm 18.86$ |
| dqn | $47.41 \pm 80.00$ | $5.11 \pm 80.00$ | $14.65 \pm 17.99$ |
| dqn+spr | $52.46 \pm 81.35$ | $6.40 \pm 81.35$ | $22.58 \pm 29.75$ |

We evaluate our method across three distinct benchmark suites: the Atari 26 subset used by Schwarzer et al. (2021) for evaluating generalization in visually complex environments, and the MinAtar (Young & Tian, 2019) and the Octax (Radji et al., 2026) suites for lightweight validation. In addition to mean and median, we also report the Inter-Quartile

Mean (IQM) of our aggregate results with a confidence interval of 95% (Agarwal et al., 2021), computed over all environment and seed runs for each algorithm. *We aim to answer three main questions*: (1) Does adding SPR improve the sample efficiency and final performance of streaming agents? (2) Does our orthogonalization strategy successfully reconcile SPR with the ObGD optimizer? (3) Does the auxiliary loss actually result in better representation quality as hypothesized?

## 5.1. Performance on Standard Environments

We compare three base algorithms, Streaming DQN (dqn), QRC($\lambda$) (qrc), and Stream Q($\lambda$) (strq), against their SPR-augmented counterparts. We use the same set of parameters as Schwarzer et al. (2021) with additional justification in Table 5. We integrate this framework into our three streaming baselines, as detailed in Sec. 4.1. Everywhere in the results, spr, orth, and orth$^2$ indicate the application of SPR, orthogonal SPR gradient projection (Sec. 4.2), and orthogonal SPR gradient projection with respect to ObGD gradients after orth (Sec. 4.3) respectively.

Table 1 presents the main results of our work. We see that QRC($\lambda$) with SPR and streaming DQN with SPR consistently outperforms the base algorithm over all environment suites and metrics, with the addition of orth providing even higher returns. The learning curves in Figure 3 corroborate this, showing the higher sampling efficiency of SPR-augmented agents through faster learning in the early stages of training. However, as hypothesized in Section 4.3, naively adding SPR to Stream Q($\lambda$) leads to performance degradation due to gradient conflicts (see Fig. 4 and Table 1). However, applying our orth$^2$ strategy successfully recovers performance, surpassing the baseline Stream Q($\lambda$) on all metrics at 100M frames on Atari.

We note that QRC($\lambda$) performs relatively worse on Atari compared to MinAtar and Octax, even though we use the same set of hyperparameters for all experiments (Appendix A.1). A breakdown of per-environment results are present in Tables 9, 10, 11 and 12 in Appendix B. A key constraint of the streaming setting is resource efficiency. Despite the addition of the dynamics model and projection heads, our method still remains computationally feasible (see Appendix A.9).

## 5.2. Latent Space Analysis

To understand *why* SPR improves performance, we analyze the structure of the learned representations. Table 2 shows the effective rank (Roy & Vetterli, 2007) of the encoder $f_\theta$'s latent space representation $z_t$ on a batch of five thousand observations. Effective rank quantifies the intrinsic dimensionality or "complexity" of the representations the encoder learns. A higher effective rank indicates that the encoder is

*Table 2.* The effective ranks of the latents computed using the encoders trained with and without SPR loss at 40M frames, averaged over all Atari 26 environments. Note that we used `orth`$^2$ for Stream Q($\lambda$) and `orth` for QRC($\lambda$) for the SPR numbers.

| Algorithm | Without SPR | With SPR (ours) |
|---|---|---|
| QRC($\lambda$) | $42.7 \pm 33.7$ | $\mathbf{1256.9 \pm 296.5}$ |
| Stream Q($\lambda$) | $466.4 \pm 120.4$ | $\mathbf{912.5 \pm 209.5}$ |
| Streaming DQN | $518.7 \pm 259.4$ | $526.7 \pm 243.4$ |

utilizing more of its capacity and capturing a richer diversity of features. We see that across the base algorithms, having the SPR loss leads to a higher effective rank.

Furthermore, the t-SNE visualization of the same observation trajectory (Fig. 2, and Fig. 9 in Appendix H) reveals a striking qualitative difference. The baseline encoder produces a scattered latent map with little structure. The SPR-augmented encoder, however, produces several continuous trajectories where temporally adjacent states are mapped to spatially adjacent points in the latent space (visualized by the color gradient).

Following Anand et al. (2019), we freeze the Atari encoders and train linear probes to predict individual ALE RAM bytes, which capture game variables such as object positions, scores, timers, and lives. Table 3 reports the average held-out $R^2$ across non-constant bytes and the number of well-predicted bytes under a fixed probe threshold. Across all tested games, QRC($\lambda$)+SPR+`orth` attains both higher per-byte $R^2$ and more well-predicted bytes than QRC($\lambda$). Zhang et al. (2024) show linear probing correlates strongly with downstream RL performance on Atari, supporting the hypothesis that better representations lead to better Q-networks and task performance.

*Table 3.* Linear probes from frozen Atari encoders to RAM bytes. Higher per-byte $R^2$ and more well-predicted bytes indicate that the latent representation preserves more game-state information.

| Game | Method | Per-byte $R^2$ | Good bytes |
|---|---|---|---|
| Alien | QRC+SPR+`orth` | **0.28** | **24/94** |
| Alien | QRC | 0.24 | 19/94 |
| Assault | QRC+SPR+`orth` | **0.30** | **24/77** |
| Assault | QRC | 0.24 | 15/77 |
| Pong | QRC+SPR+`orth` | **0.55** | **17/28** |
| Pong | QRC | 0.49 | 13/28 |

### 5.3. Comparison with PCGrad

A natural alternative to `orth`$^2$ is PCGrad (Yu et al., 2020), which projects gradients when two objectives conflict. However, PCGrad treats the objectives symmetrically: when $g_{RL} \cdot g_{SPR} < 0$, it modifies the RL gradient as $\hat{g}_{RL} = g_{RL} - (g_{RL} \cdot g_{SPR}/\|g_{SPR}\|^2)g_{SPR}$, and applies the analogous projection to $g_{SPR}$. This symmetry is undesirable for Stream Q($\lambda$), because ObGD's adaptive step size is calibrated for the original RL update direction. Deflecting that direction can break the local stability logic of Eq. 1. Our `orth`$^2$ update is instead asymmetric: it leaves the RL/ObGD update untouched and constrains only the auxiliary SPR update to the orthogonal complement.

*Table 4.* PCGrad comparison for Stream Q($\lambda$)+SPR on Atari at 40M frames with three seeds, reported as human-normalized scores. StreamQ denotes the no-SPR baseline.

| Game | PCGrad | `orth`$^2$ (ours) | StreamQ |
|---|---|---|---|
| Alien | $0.10 \pm 0.01$ | $0.10 \pm 0.01$ | $0.06 \pm 0.02$ |
| Breakout | $0.77 \pm 0.02$ | $\mathbf{1.46 \pm 0.17}$ | $1.43 \pm 0.08$ |
| MsPacman | $0.13 \pm 0.02$ | $\mathbf{0.15 \pm 0.01}$ | $0.11 \pm 0.02$ |
| Pong | $0.35 \pm 0.28$ | $\mathbf{0.71 \pm 0.15}$ | $0.17 \pm 0.09$ |
| Seaquest | $0.01 \pm 0.00$ | $0.02 \pm 0.00$ | $0.02 \pm 0.01$ |

Table 4 confirms this distinction empirically. PCGrad helps over the no-SPR baseline on Alien, MsPacman, and Pong, but substantially underperforms `orth`$^2$ on Breakout and Pong and falls below the no-SPR StreamQ baseline on Breakout and Seaquest. In contrast, `orth`$^2$ matches or improves over StreamQ on all five games. These results support treating RL as the primary objective and SPR as an auxiliary objective when the policy update is controlled by an adaptive optimizer such as ObGD.

### 5.4. Ablations

To justify our choice of hyperparameters, namely $K$=5, $\lambda_{\text{SPR}}$=2, and image augmentations with EMA $\tau$=0, we conduct experiments with several values of these parameters (Table 5). We observe the same trends as those presented in Schwarzer et al. (2021). The case with $\tau$=0 is particularly interesting since it means that the target networks $f_{\theta'}$ and $P_{\phi'}$ are direct copies of the online networks $f_\theta$ and $P_\phi$ respectively, and thus image augmentations are required as a form of regularization.

For streaming DQN, we also conduct an experiment with different replay buffer (`rb`) sizes, `rb`=1 being the default. A replay buffer of size 5 does result in marginally better returns, however it still falls short of SPR, which also stores 5 states ($K$=5). This indicates that the gains associated with SPR are not due to the stored parameters alone.

We also conducted an experiment using the CURL framework (Laskin et al., 2020), which employs a contrastive loss as an auxiliary representation objective and used a replay buffer of size 5 for a fair comparison with SPR (see Appendix F). However, contrastive methods based on negative examples famously require very large batch sizes, and as such are exceedingly stream-unfriendly. Our results confirm this as our CURL-based QRC($\lambda$) agent fails to learn anything.

*Table 5.* Ablations on the Atari 26 environments at 40M frames and five random seeds (Human Normalized). We evaluate the effect on performance of the SPR prediction depth $K$, the SPR update weight $\lambda_{SPR}$, and the EMA weight $\tau$ for the SPR target networks. We also look at different sizes of replay buffers for streaming DQN. Finally, we assess the quality of the CURL framework for representation learning with QRC($\lambda$).

| Algorithm | Mean Return | Median Return | IQM Return |
|---|---|---|---|
| *Effect of prediction depth $K$* | | | |
| qrc+spr K=5 | $1.13 \pm 1.19$ | $0.46 \pm 1.18$ | $0.61 \pm 0.39$ |
| qrc+spr K=3 | $0.96 \pm 1.07$ | $0.49 \pm 1.07$ | $0.49 \pm 0.26$ |
| qrc+spr+K=1 | $0.82 \pm 0.93$ | $0.31 \pm 0.93$ | $0.46 \pm 0.33$ |
| *Effect of SPR weight $\lambda_{SPR}$* | | | |
| qrc+spr $\lambda$=2.0 | $1.13 \pm 1.19$ | $0.46 \pm 1.18$ | $0.61 \pm 0.39$ |
| qrc+spr $\lambda$=1.0 | $1.09 \pm 1.25$ | $0.43 \pm 1.25$ | $0.60 \pm 0.45$ |
| qrc+spr $\lambda$=0.5 | $1.18 \pm 1.41$ | $0.45 \pm 1.41$ | $0.55 \pm 0.37$ |
| *Effect of Augmentation and EMA $\tau$ ($\tau_1 = 0$ and $\tau_2 = 0.99$)* | | | |
| qrc+spr + Aug, $\tau_1$ | $1.13 \pm 1.19$ | $0.46 \pm 1.18$ | $0.61 \pm 0.39$ |
| qrc+spr, $\tau_2$ | $1.06 \pm 1.19$ | $0.46 \pm 1.19$ | $0.60 \pm 0.39$ |
| *Effect of Replay Buffer* | | | |
| dqn+spr | $0.35 \pm 0.53$ | $0.19 \pm 0.53$ | $0.22 \pm 0.11$ |
| dqn rb=5 | $0.30 \pm 0.51$ | $0.11 \pm 0.50$ | $0.13 \pm 0.07$ |
| dqn rb=1 | $0.29 \pm 0.52$ | $0.10 \pm 0.52$ | $0.12 \pm 0.08$ |
| *Effect of Representation Learning Frameworks* | | | |
| qrc+spr | $1.13 \pm 1.19$ | $0.46 \pm 1.18$ | $0.61 \pm 0.39$ |
| qrc+CURL | $-0.03 \pm 0.05$ | $-0.01 \pm 0.00$ | $-0.01 \pm 0.01$ |

# 6. Discussion

Our results highlight a fundamental inefficiency in current streaming RL methods: the discarding of transitions after a single update leaves structural information unexploited. By integrating Self-Predictive Representations (SPR), we force the encoder to extract the dynamics of the environment from the data stream before it is lost. This approach trades a degree of computational speed for significantly improved sample efficiency. While the auxiliary task introduces additional networks and a small state buffer, the gain in representation quality justifies the overhead.

Our results also suggest that explicit representation learning can *bridge* a significant portion of the gap caused by the absence of replay memory. By replaying transitions virtually through the predictive model, the agent captures long-term dependencies that a pure TD-error signal misses.

**The Necessity of Orthogonalization**    The failure of naive SPR integration with Stream Q($\lambda$) and its rescue by our `orth`[2] update highlight a key lesson for streaming learning: when an optimizer like ObGD aggressively adapts step sizes to stabilize the RL loss, auxiliary gradients must be tightly controlled to prevent destructive interference. Orthogonalization provides a geometric safeguard, enabling the encoder to learn useful representations without compromising the fragile stability of the policy update in the streaming setting.

**Limitations and Future Directions**    At present, we do not make use of the latent dynamics model learned by our streaming version of SPR. An interesting follow-up would be to apply deep model-based Reinforcement Learning methods to the streaming context. Investigating whether new learning signals derived from planning or world-modeling can improve the overall stability and sample efficiency of the training process remains a promising avenue for future research. Another promising direction is to evaluate SPR on continuous control policies in a streaming setting.

An interesting study would be to analyze whether learning the transition dynamics reduces agents' ability to quickly adapt to environmental changes compared to base streaming agents. We also designed our method to be efficient enough for CPU training, however, the addition of the SPR objective introduces a non-trivial computational cost, increasing the wall-clock time per step. Future work could explore sparse application of the auxiliary loss (e.g., updating the SPR module only every k steps) to balance runtime and representation quality.

A notable challenge lies in reconciling SPR with the ObGD optimizer in Stream Q($\lambda$), where a performance gap persists compared to SGD-based baselines despite our orthogonalization fix, future work could investigate theoretical formulations of ObGD that naturally accommodate auxiliary objectives.

# 7. Conclusion

We integrated Self-Predictive Representations into the streaming RL pipeline through an orthogonal gradient update scheme that stabilizes representation learning against highly temporally correlated streaming data. We further resolve gradient conflicts arising from the simultaneous application of ObGD for the Reinforcement Learning objective and SGD for the SPR auxiliary loss, ensuring that representation learning does not interfere with the policy updates induced by ObGD. This approach systematically outperforms current streaming baselines across the Atari, MinAtar, and Octax benchmarks and learns high-rank latent spaces without the aid and memory overhead of large replay buffers. Our results establish that dense auxiliary supervision, rather than just improved value estimation, is a highly effective strategy for deploying sample-efficient RL agents on constrained hardware where every observation must be fully exploited before it is discarded.

# Acknowledgements

Nilaksh is partly supported by a grant (DOI 2009238) from the Fonds de recherche du Québec (FRQNT). Sarath Chan-

dar is supported by the Canada CIFAR AI Chairs program, the Canada Research Chair in Lifelong Machine Learning, and the NSERC Discovery Grant.

This research was enabled in part by compute resources provided by Mila (mila.quebec) and the Digital Research Alliance of Canada (alliancecan.ca).

## Impact Statement

This paper presents work aimed at advancing the field of Machine Learning, specifically in the domain of streaming and on-device Reinforcement Learning (RL). We believe our contributions have several societal implications worth highlighting.

First, the shift towards on-device learning offers significant *privacy* advantages. By processing transitions sequentially and discarding them immediately, our approach minimizes data retention and eliminates the need to transmit raw observations to remote servers.

Second, regarding *energy consumption*, our focus on lightweight, memory-efficient algorithms (avoiding large batches and extensive memory storage) aligns with the goal of reducing carbon emissions. By reducing the computational and memory footprint required for effective RL, we facilitate deployment on low-power edge devices.

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

# A. Experimental details

This section goes over the hyperparameter choices, architectural choices and implementation details in more depth to allow for better understanding of our setup and easier reproduction of our results.

## A.1. Hyperparameters

We provide the details of all our hyperparameter choices in Table 6. For the image augmentations, we use Intensity shift ($\sigma = 0.05$) and Random shifts ($\pm 4$ pixels).

*Table 6.* Hyperparameters across streaming baselines and our proposed QRC+SPR setting.

| Parameter | Streaming DQN | StreamQ ($\lambda$) | QRC ($\lambda$) | + SPR |
|---|---|---|---|---|
| Optimizer | SGD | OBGD ($\kappa = 2.0$) | SGD | SGD + Ortho ($\beta_{orth} = 0.99$) |
| Learning rate ($\alpha$) | $10^{-4}$ | 1.0 | $10^{-4}$ | $10^{-4}$ |
| Batch size | 1 | 1 | 1 | 1 |
| Replay buffer size | 1 | (N/A) | (N/A) | (N/A) |
| Q-target network | True | False | False | False |
| $\lambda$-return | 0 ($n = 1$) | 0.8 | 0.8 | 0.8 |
| Discount Factor | 0.99 | 0.99 | 0.99 | 0.99 |
| $\epsilon$ Explore Fraction | 0.1 | 0.2 | 0.1 | same as base algo |
| Reward Norm. | True | True | True | True |
| Observation Norm. | True | True | True | True |
| *Architecture Details* | | | | |
| Activation | Leaky ReLU | Leaky ReLU | Leaky ReLU | Leaky ReLU |
| Layer Norm | True | True | True | True |
| Sparse Init | 0.9 | 0.9 | 0.9 | 0.9 |
| *QRC Specific* | | | | |
| Auxiliary LR scale | N/A | N/A | 0.1 | 0.1 |
| Reg. coefficient ($\beta_{qrc}$) | N/A | N/A | 1.0 | 1.0 |
| *SPR Specific* | | | | |
| SPR weight ($\lambda_{SPR}$) | N/A | N/A | N/A | 2.0 |
| Prediction depth ($K$) | N/A | N/A | N/A | 5 |
| Target Networks EMA ($\tau$) | N/A | N/A | N/A | 0.0 |
| Shared Proj. Head | N/A | N/A | N/A | True |
| Data Augmentation | N/A | N/A | N/A | True |

## A.2. Atari environments

All agents evaluated with our SPR settings are trained on environments from the Arcade Learning Environment (ALE) (Bellemare et al., 2013). We take the set of 26 environments used by Schwarzer et al. (2021). The individual environment returns at 40M frames and 100M frames are given in Table 11 and 12 respectively. This shows that the trends hold at even higher training frames. In order to normalize the returns, we take human and random policy data from Badia et al. (2020), and obtain the normalized return as: norm score = (score - random) / (human - random).

We use the same setup as Elsayed et al. (2024) regarding the Atari environments. Each environment frame is down-sampled to a shape of $84 \times 84$ and converted to grayscale. Frames are then stacked by 4 (to deal with partial observability) and this constitutes the (unnormalized) state returned by the environment. At the beginning of each episode, the agent is forced to take a random number of no-op actions (up to 30). For environments where the game starts after the firing action has been taken, the agent is forced to take a random action at the start. Episodes are terminated when the agent loses all of its lives and each action taken by the agent is repeated 4 times. Finally, we normalize the observations and rewards before giving them as input to the agent, according to Elsayed et al. (2024).

## A.3. MinAtar Environments

The MinAtar environments (Young & Tian, 2019) offer a simplified version of Atari with more focus on the behavior learning aspect of solving the games by simplifying the representation complexity. The frames are $10 \times 10 \times n$ in size where

each of the $n$ channels capture an element of the game dynamics. They also encode temporal information by displaying a trail behind some objects. We evaluate on all five implemented environments (see Table 9).

### A.4. Octax Environments

Octax (Radji et al., 2026) is a recent suite that provides high-performance arcade environments through JAX-based CHIP-8 emulation. These games serve as a GPU-accelerated alternative to the Atari benchmark, offering easier cognitive demands while enabling massive parallelization. The observation space consists of raw $32 \times 64$ monochrome frames, which we provide to the agent as a $(4, 32, 64)$ boolean array to incorporate temporal information via a frame skip of 4. Rewards and termination conditions are extracted directly from game-specific memory registers and internal logic, ensuring perfect fidelity to the original mechanics. We evaluate our agents across 8 games spanning the puzzle, action, and strategy genres. Octax games have a binary observation space with simpler environments compared to Atari, and thus serve as another validation suit along with MinAtar.

### A.5. LayerNorm

It is important to note that throughout this paper, when we mention using layer normalization we refer to the parameter-free layer normalization used by Elsayed et al. (2024). Meaning the LayerNorm we use does not have the usual two learnable parameters of re-scaling and bias. For a given input $\mathbf{x} \in \mathbb{R}^n$, it corresponds to the following formula:

$$\forall i \in [\![1, n]\!], \ \text{LAYERNORM}(\mathbf{x})_i = \frac{\mathbf{x}_i - \mu}{\sqrt{\sigma^2 + \epsilon}}, \qquad \text{where } \mu = \frac{1}{n}\sum_{k=1}^{n}\mathbf{x}_k \text{ and } \sigma^2 = \frac{1}{n}\sum_{k=1}^{n}(\mathbf{x}_k - \mu)^2 \tag{5}$$

Epsilon is a small quantity added for numerical stability and in our case we use the default pytorch value of $\epsilon = 10^{-5}$.

### A.6. Architecture design (Q-network)

*Table 7.* The network architectures used for each of the environments. Here $|\mathcal{A}|$ is the number of actions.

|  |  | **Atari** | **Octax** | **MinAtar** |
|---|---|---|---|---|
| Training Frames |  | 100M / 40M | 5M | 20M |
| Q head layers |  | $(512, |\mathcal{A}|)$ | $(512, |\mathcal{A}|)$ | $(128, |\mathcal{A}|)$ |
| Encoder | Channel List | 32, 64, 64 | 32, 64, 64 | 16 |
|  | Kernel List | 8×8, 4×4, 2×2 | 4×8, 4×4, 2×2 | 3×3 |
|  | Strides List | 4×4, 2×2, 1×1 | 2×4, 2×2, 1×1 | 1×1 |
| Dynamics Net | Channel List | 64, 64 | 64, 64 | 16, 16 |
|  | Kernel List | 3×3, 3×3 | 3×3, 3×3 | 3×3, 3×3 |
|  | Strides List | 1×1, 1×1 | 1×1, 1×1 | 1×1, 1×1 |
| Projection Net layers |  | (512, ) | (512, ) | (128, ) |
| Prediction Net layers |  | (512, ) | (512, ) | (128, ) |

The network specifications for each of the environments are given in Table 7. The networks relevant for Q-learning are the encoder and the Q head. We use the default encoder architecture for Atari consisting of three convolutional layers. For Octax this is slightly modified as it has a $32 \times 64$ observation spatial dimension. Thus we use a $4 \times 8$ kernel for the first layer. For MinAtar, we follow Elelimy et al. (2025) and use just one convolution layer. The output of the first fully-connected layer is passed through a LayerNorm and a LeakyReLU(0.01) activation. The last layer has no bias term and directly outputs the Q-values. We also apply the same sparse initialization scheme as Elsayed et al. (2024) to all the layers used in the Q-network (encoder and fully-connected layers). This effectively sets all biases to zero and initializes 90% of each layer's weights to zero while the 10% left follows the LeCun initialization scheme (LeCun et al., 2012).

### A.7. Architecture design (SPR-network)

The SPR part of the network shares the encoder with the Q-learning part of the network, and uses the remaining components given in Table 7. A difference between our architecture and the one from Schwarzer et al. (2021) is that we use layer normalization before each activation (as opposed to batch normalization after some of the layers). For this reason, we drop the renormalization of the encoder and dynamics network outputs, as we already use LayerNorm everywhere. The dynamics

network $D_\psi$ consists of two convolutional layers, both using kernel sizes of $3 \times 3$, padding of type `"same"` and padding mode of type `"reflect"`. The first convolutional layer also takes a one-hot encoding of the action as input, concatenated along the channel dimension. Again, all pre-activations are passed through a LayerNorm and a LeakyReLU(0.01) activation. Then follows the prediction layer $P_\phi$. By default, this layer is shared with the Q-network. As such, it is exactly the same as the first fully-connected layer of the Q-network head. Schwarzer et al. (2021) justify this decision by arguing that using the first layer of the Q-network head allows the SPR objective to affect far more of the Q-network. In the case of the `orth`² update scheme (Sec. 4.3), we also orthogonalize the SPR gradients with respect to the ObGD gradients for this shared prediction layer. Finally the projection $q_\omega$ layer is a fully-connected layer. None of these two last layers use LayerNorm or activation functions. The predicted next latents are iteratively computed via the dynamics model: $\hat{z}_{t+j+1} = D_\psi(\hat{z}_{t+j}, a_{t+j})$ for $0 \leq j < K$ and $\hat{z}_t = z_t$. These prediction are then passed through the prediction and projection layers. Their targets are computed from the encoder directly applied to the next states in the trajectory, followed by the prediction layer and no gradients are propagated through these operations.

## A.8. Optimizer

As mentioned in the method section, because Stream-Q with the ObGD optimizer uses eligibility traces-based updates and SPR uses SGD-based updates, weight sharing could not be implemented as easily as back-propagating the sum of the two losses directly. The update process of non-shared parameters is unchanged and is performed using the gradients or eligibility traces deriving from the relevant loss. For shared parameters, we perform two separate phases of back-propagation (one per loss). We compute the updates induced by each one of them, either with eligibility traces or SGD, and apply the final update to each shared parameter as the weighted sum of the updates induced by the SPR objective and the Q-learning objective. The optimal learning rate computation in the ObGD optimizer specifically only considers the $L_1$ norm of the update coming from the Q-learning objective and disregards any update coming from the SPR objective. This overestimation of the maximum stable learning rate should however be accounted for by the $\kappa$ coefficient, taken to be equal to 2 as in Elsayed et al. (2024), as well as our mixing coefficient $\mu_{shared} = 0.5$.

## A.9. Computational resources

*Table 8.* Experiment times, trainable parameter count and memory usage on Atari for 40M frames using 4 CPU cores.

| Algorithm | Time (hours) | Approx. Parameters | Process Memory Usage |
|---|---|---|---|
| dqn+spr | $31.25 \pm 3.72$ | 1.94M | $1.11 \pm 0.03$ GB |
| dqn | $20.73 \pm 1.21$ | 1.68M | $1.03 \pm 0.02$ GB |
| strq+spr+orth | $25.27 \pm 0.34$ | 1.94M | $1.35 \pm 0.09$ GB |
| strq+spr | $38.90 \pm 1.32$ | 1.94M | $1.34 \pm 0.06$ GB |
| strq | $19.52 \pm 2.70$ | 1.68M | $1.13 \pm 0.01$ GB |
| qrc+spr+orth | $48.66 \pm 3.38$ | 3.71M | $1.59 \pm 0.09$ GB |
| qrc+spr | $51.16 \pm 0.51$ | 3.71M | $1.49 \pm 0.07$ GB |
| qrc | $36.85 \pm 3.90$ | 3.37M | $1.34 \pm 0.06$ GB |

All experiments were conducted using standard CPUs with limited cores, validating that our approach is suitable for on-device learning scenarios where GPU acceleration may not be available. To run these experiments (both with and without SPR), we used 4 CPU cores with 4GB RAM. The networks and the training code was written in JAX. As shown in Table 8, the streaming DQN and Stream Q($\lambda$) agents completed 40M frames of training in roughly 20 to 35 hours, while the QRC($\lambda$) variants took longer, taking about 35 to 50 hours. In the original SPR codebase, 400k frames experiments take from 2 to 6 hours to complete depending on the algorithm. We also list the total trainable parameter size for each algorithm.

## A.10. SPR and Orthogonal Updates Algorithm

---

**Algorithm 1** Streaming Self-Predictive Representations with Orthogonal Updates

---

1: **Input:** Stream of transitions
2: **Parameters:** $\theta$ (online enc), $\theta'$ (target enc), $\psi$ (transition), $\phi$ (online proj), $\phi'$ (target proj), $\omega$ (predictor), $\xi$ (Q-head)
3: **Config:** Flags `orth`, `orth`$^2$
4: Denote prediction horizon as $K$
5: Initialize momentum history $m \leftarrow 0$ for all SPR modules $\eta \in \{\theta, \psi, \phi, \omega\}$
6:
7: **for** step $t$ in training **do**
8:     Collect experience $(s_t, a_t, r_t, s_{t+1})$
9:     Store transition in temporary buffer
10:     **if** buffer size $< K$ **then**
11:         **continue**
12:     **end if**
13:     {SPR Phase: Recompute latents from past states}
14:     Retrieve trajectory $s_{t-K+1}, \ldots, s_{t+1}$ and actions $a_{t-K+1}, \ldots, a_t$
15:     **if** augmentation **then**
16:         $s \leftarrow \text{augment}(s)$ for all $s$ in trajectory
17:     **end if**
18:     $z \leftarrow f_\theta(s_{t-K+1})$ {Initial online latent}
19:     Initialize SPR loss $l_{SPR} \leftarrow 0$
20:     **for** $k = 0$ **to** $K - 1$ **do**
21:         $\hat{z}_{next} \leftarrow D_\psi(z, a_{t-K+1+k})$ {Transition model}
22:         $\tilde{z}_{next} \leftarrow f_{\theta'}(s_{t-K+1+k+1})$ {Target representations}
23:         $\hat{y} \leftarrow q_\omega(P_\phi(\hat{z}_{next}))$ {Online projection & prediction}
24:         $\tilde{y} \leftarrow P_{\phi'}(\tilde{z}_{next})$ {Target projection}
25:         $l_{SPR} \leftarrow l_{SPR} - \left( \frac{\hat{y}}{\|\hat{y}\|_2} \cdot \frac{\text{SG}[\tilde{y}]}{\|\text{SG}[\tilde{y}]\|_2} \right)$ {Cosine similarity}
26:         $z \leftarrow \hat{z}_{next}$
27:     **end for**
28:     $\delta_{SPR} \leftarrow \nabla_{\theta,\psi,\phi,\omega}(\lambda_{SPR} l_{SPR})$ {Compute SPR gradients}
29:     {Decorrelating Updates (Sec. 4.2)}
30:     **if** `orth` **then**
31:         **for** each module param $\eta \in \{\theta, \psi, \phi, \omega\}$ **do**
32:             $\delta_\eta \leftarrow \delta_\eta - \text{proj}_{m_\eta}(\delta_\eta)$ {Orthogonalize current grad against history}
33:             $m_\eta \leftarrow \beta m_\eta + (1 - \beta)\delta_\eta$ {Update history}
34:         **end for**
35:     **end if**
36:     {RL Phase & Conflict Resolution (Sec. 4.3)}
37:     $u^{RL} \leftarrow \text{ObGDVector}(s_t, a_t, r_t, s_{t+1}; \theta, \xi)$ {Calculates updates for shared enc and Q-head}
38:     **if** `orth`$^2$ **and** using ObGD **then**
39:         **for** shared params $\theta$ **do**
40:             $\delta_\theta \leftarrow \delta_\theta - \text{proj}_{u_\theta^{RL}}(\delta_\theta)$ {Project SPR grad away from RL update}
41:         **end for**
42:     **end if**
43:     {Apply combined updates}
44:     $\theta \leftarrow \theta + u_\theta^{RL} - \alpha\delta_\theta$ {Update shared encoder}
45:     $\xi \leftarrow \xi + u_\xi^{RL}$ {Update Q-head (RL only)}
46:     $\psi, \phi, \omega \leftarrow \text{optimizer}(\psi, \phi, \omega; -\alpha\delta_{SPR})$ {Update SPR modules}
47:
48:     $\theta' \leftarrow (1 - \tau)\theta + \tau\theta'$ {Update target encoder (EMA)}
49:     $\phi' \leftarrow (1 - \tau)\phi + \tau\phi'$ {Update target projection (EMA)}
50: **end for**

---

# B. Additional Results

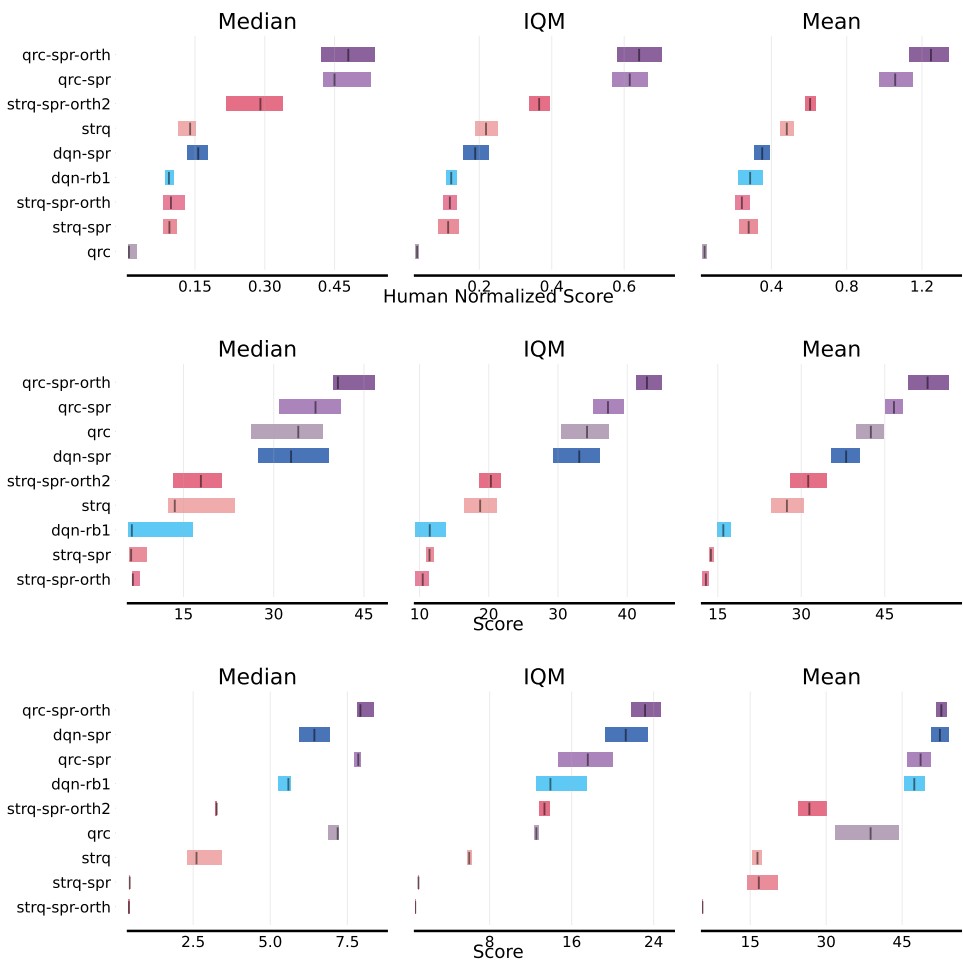

*Figure 5.* The aggregate metrics with a 95% confidence interval for the algorithms across different domains: Atari (`TOP`), MinAtar (`MIDDLE`) and Octax (`BOTTOM`). Here IQM stands for interquartile mean. Each experiment was run for five random seeds. We see that adding SPR increases aggregate performance for both streaming DQN and QRC(λ) across all metrics. The addition of orthogonal updates also helps SPR work better in the streaming setting. SPR fails to work with Stream Q(λ) due to gradient conflicts. However after orthogonalizing SPR gradients with respect to ObGD gradients as described in Sec. 4.3, we see a marked improvement in SPR performance, with a higher median and IQM than Stream Q(λ).

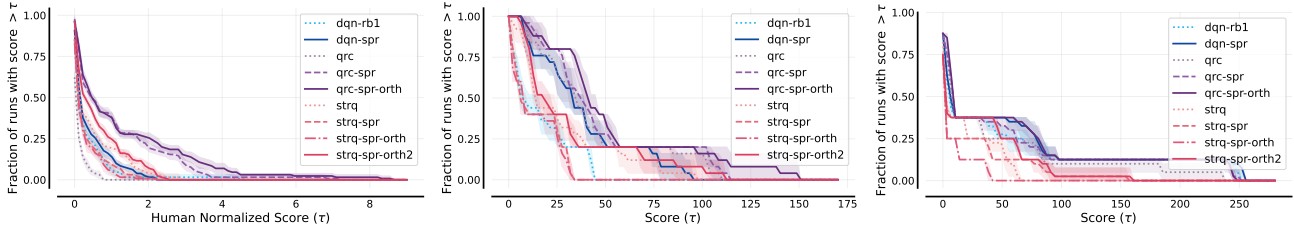

*Figure 6.* The performance profiles of the different algorithms across the Atari, MinAtar and Octax games, displaying their probability of achieving a certain score. We see that for QRC(λ) and DQN adding SPR loss leads to an increase in performance across scores. For Stream Q(λ), the `orth²` method turns out to be better or on par with the base algorithm

## B.1. MinAtar Mean Scores Per Environment

*Table 9.* MinAtar mean scores after training for 5M frames

| Environment | dqn-rb1 | dqn-spr | qrc | qrc-spr | qrc-spr-orth | strq | strq-spr | strq-spr-orth | strq-spr-orth2 |
|---|---|---|---|---|---|---|---|---|---|
| Asterix-v1 | 6.85 ± 5.57 | 28.36 ± 3.70 | 28.73 ± 8.72 | 35.75 ± 4.22 | **37.21 ± 4.89** | 15.29 ± 4.25 | 3.22 ± 1.06 | 1.83 ± 0.59 | 15.83 ± 5.00 |
| Breakout-v1 | 6.06 ± 0.25 | 9.42 ± 2.29 | 10.98 ± 2.77 | 13.19 ± 3.50 | **13.38 ± 4.77** | 11.02 ± 1.47 | 6.91 ± 1.07 | 6.87 ± 0.51 | 11.85 ± 0.92 |
| Freeway-v1 | 43.14 ± 0.93 | 43.54 ± 4.23 | 44.09 ± 4.15 | 47.66 ± 4.84 | **52.27 ± 2.59** | 31.95 ± 5.70 | 25.31 ± 1.13 | 23.63 ± 2.96 | 32.15 ± 1.40 |
| Seaquest-v1 | 0.94 ± 0.28 | 25.91 ± 10.30 | 29.01 ± 4.70 | 29.34 ± 2.77 | **39.91 ± 1.41** | 5.13 ± 2.85 | 2.15 ± 0.25 | 2.04 ± 0.23 | 9.72 ± 4.12 |
| SpaceInvaders-v1 | 22.98 ± 4.14 | 83.10 ± 8.45 | 99.71 ± 8.50 | 107.35 ± 4.07 | **120.72 ± 19.76** | 73.72 ± 14.97 | 31.21 ± 1.31 | 29.95 ± 2.03 | 86.67 ± 17.64 |
| **Aggregate** | 15.99 ± 15.46 | 38.06 ± 24.98 | 42.50 ± 30.47 | 46.66 ± 32.32 | **52.70 ± 36.27** | 27.42 ± 24.81 | 13.76 ± 12.09 | 12.86 ± 11.69 | 31.25 ± 28.81 |

## B.2. Octax Mean Scores Per Environment

*Table 10.* Octax mean scores after training for 20M frames

| Environment | dqn-rb1 | dqn-spr | qrc | qrc-spr | qrc-spr-orth | strq | strq-spr | strq-spr-orth | strq-spr-orth2 |
|---|---|---|---|---|---|---|---|---|---|
| airplane | 2.71 ± 1.40 | 3.46 ± 0.04 | 3.60 ± 0.35 | 3.72 ± 0.11 | **3.75 ± 0.11** | 0.28 ± 0.35 | 0.00 ± 0.00 | -0.02 ± 0.05 | 0.00 ± 0.00 |
| deep | 3.12 ± 0.99 | 5.27 ± 0.65 | 6.09 ± 0.38 | 7.61 ± 0.56 | **8.85 ± 0.98** | 2.54 ± 0.51 | 0.19 ± 0.01 | 0.01 ± 0.00 | 3.47 ± 0.27 |
| filter | 76.86 ± 12.75 | **80.22 ± 10.33** | 79.78 ± 6.67 | 57.02 ± 5.88 | 72.19 ± 6.92 | 52.74 ± 6.19 | 0.71 ± 0.04 | 0.69 ± 0.05 | 45.96 ± 2.42 |
| flight_runner | 247.18 ± 2.95 | **250.88 ± 3.31** | 183.85 ± 75.06 | 243.14 ± 6.63 | 243.39 ± 2.21 | 17.99 ± 0.02 | 98.88 ± 29.36 | 11.11 ± 0.01 | 102.14 ± 27.87 |
| pong | -4.54 ± 0.50 | **-2.64 ± 1.43** | -6.65 ± 0.44 | -5.04 ± 0.51 | -4.36 ± 0.33 | -3.55 ± 1.75 | -8.02 ± 0.16 | -7.95 ± 0.05 | -5.51 ± 0.04 |
| spacejam | 7.10 ± 0.35 | 7.54 ± 0.44 | **8.19 ± 0.64** | 7.83 ± 0.11 | 7.87 ± 0.05 | 3.11 ± 0.74 | 3.16 ± 0.08 | 2.11 ± 0.03 | 3.11 ± 0.11 |
| ufo | 45.67 ± 13.47 | 74.05 ± 10.88 | 32.21 ± 0.92 | 72.14 ± 23.31 | **86.63 ± 6.42** | 58.21 ± 6.14 | 38.47 ± 1.01 | 38.48 ± 0.62 | 63.45 ± 1.73 |
| worm | 1.17 ± 0.21 | 0.94 ± 0.13 | 3.15 ± 0.33 | 2.78 ± 0.39 | **3.67 ± 1.76** | 0.14 ± 0.03 | 0.18 ± 0.02 | 0.17 ± 0.02 | 0.85 ± 0.06 |
| **Aggregate** | 47.41 ± 80.00 | 52.46 ± 81.35 | 38.78 ± 60.56 | 48.65 ± 78.10 | **52.75 ± 78.96** | 16.43 ± 23.35 | 16.70 ± 33.73 | 5.57 ± 13.34 | 26.68 ± 36.96 |

## B.3. Atari Mean Scores Per Environment

*Table 11.* Atari mean scores after training for 40M frames (Human Normalized)

| Environment | dqn | dqn+spr | qrc | qrc+spr | qrc+spr+orth | strq | strq+spr | strq+spr+orth | strq+spr+orth$^2$ |
|---|---|---|---|---|---|---|---|---|---|
| Alien | 0.08 ± 0.01 | 0.07 ± 0.01 | -0.01 ± 0.00 | 0.13 ± 0.01 | **0.15 ± 0.01** | 0.06 ± 0.02 | 0.05 ± 0.05 | 0.03 ± 0.04 | 0.10 ± 0.01 |
| Amidar | 0.09 ± 0.00 | 0.09 ± 0.03 | 0.03 ± 0.00 | 0.17 ± 0.01 | **0.18 ± 0.01** | 0.14 ± 0.02 | 0.09 ± 0.03 | 0.07 ± 0.02 | 0.07 ± 0.01 |
| Assault | 0.78 ± 0.16 | 0.30 ± 0.24 | 0.21 ± 0.01 | 3.13 ± 0.33 | **3.26 ± 0.32** | 1.48 ± 0.42 | 0.97 ± 0.19 | 0.90 ± 0.24 | 1.18 ± 0.00 |
| Asterix | 0.09 ± 0.01 | 0.07 ± 0.02 | 0.02 ± 0.01 | 0.16 ± 0.03 | **0.19 ± 0.01** | 0.09 ± 0.01 | 0.03 ± 0.03 | 0.03 ± 0.01 | 0.11 ± 0.04 |
| BankHeist | -0.01 ± 0.00 | 0.02 ± 0.01 | -0.00 ± 0.00 | 0.08 ± 0.05 | **0.14 ± 0.18** | -0.00 ± 0.01 | 0.01 ± 0.02 | 0.01 ± 0.02 | 0.02 ± 0.01 |
| BattleZone | 1.09 ± 0.42 | 1.85 ± 0.31 | 0.12 ± 0.10 | 1.97 ± 0.21 | **2.16 ± 0.24** | 0.13 ± 0.01 | 0.64 ± 0.65 | 0.48 ± 0.60 | 1.86 ± 0.30 |
| Boxing | 0.17 ± 0.10 | 0.32 ± 0.15 | -0.15 ± 0.00 | 0.60 ± 0.08 | **0.61 ± 0.02** | -0.15 ± 0.01 | 0.04 ± 0.22 | 0.01 ± 0.19 | 0.46 ± 0.08 |
| Breakout | 0.71 ± 0.11 | 1.07 ± 0.11 | 0.06 ± 0.00 | 2.74 ± 0.20 | **3.81 ± 0.56** | 1.43 ± 0.08 | 0.97 ± 0.62 | 0.72 ± 0.35 | 1.46 ± 0.17 |
| ChopperCommand | 0.03 ± 0.01 | 0.13 ± 0.04 | 0.03 ± 0.00 | **0.23 ± 0.08** | 0.13 ± 0.16 | 0.10 ± 0.05 | 0.06 ± 0.06 | 0.12 ± 0.07 | 0.09 ± 0.03 |
| CrazyClimber | 0.94 ± 0.07 | 1.63 ± 0.34 | 0.33 ± 0.02 | 2.97 ± 0.48 | **3.52 ± 0.33** | 1.56 ± 0.33 | 0.98 ± 0.45 | 0.75 ± 0.26 | 2.15 ± 0.21 |
| DemonAttack | 0.02 ± 0.02 | **0.37 ± 0.03** | 0.08 ± 0.02 | 0.31 ± 0.03 | 0.34 ± 0.04 | 0.07 ± 0.02 | 0.09 ± 0.02 | 0.09 ± 0.01 | 0.20 ± 0.01 |
| Freeway | 0.00 ± 0.00 | 0.57 ± 0.46 | 0.56 ± 0.28 | **1.02 ± 0.02** | 0.83 ± 0.42 | 0.40 ± 0.49 | 0.44 ± 0.36 | 0.73 ± 0.02 | 0.87 ± 0.04 |
| Frostbite | 0.06 ± 0.02 | 0.05 ± 0.01 | 0.01 ± 0.01 | 0.06 ± 0.03 | **0.14 ± 0.12** | 0.12 ± 0.12 | 0.08 ± 0.04 | 0.05 ± 0.01 | 0.06 ± 0.02 |
| Gopher | 0.11 ± 0.05 | 0.11 ± 0.13 | -0.05 ± 0.00 | **0.77 ± 0.17** | 0.53 ± 0.30 | 0.38 ± 0.27 | 0.02 ± 0.06 | 0.01 ± 0.08 | 0.15 ± 0.08 |
| Hero | 0.04 ± 0.08 | 0.17 ± 0.02 | -0.02 ± 0.01 | 0.23 ± 0.09 | **0.39 ± 0.01** | 0.22 ± 0.12 | 0.13 ± 0.19 | 0.07 ± 0.12 | 0.29 ± 0.11 |
| Jamesbond | 0.37 ± 0.05 | 0.76 ± 0.69 | 0.45 ± 0.16 | 1.43 ± 0.57 | **1.69 ± 0.47** | 0.56 ± 0.07 | 0.13 ± 0.12 | 0.21 ± 0.11 | 0.34 ± 0.28 |
| Kangaroo | 2.15 ± 1.86 | 0.69 ± 0.13 | 0.36 ± 0.07 | 4.06 ± 2.48 | **6.17 ± 2.79** | 0.57 ± 0.08 | 0.47 ± 0.13 | 0.42 ± 0.08 | 1.02 ± 0.75 |
| Krull | -0.72 ± 0.37 | -0.91 ± 0.55 | -1.09 ± 0.04 | 3.17 ± 0.27 | **3.28 ± 0.71** | 2.18 ± 0.43 | 1.09 ± 0.81 | 0.81 ± 0.73 | 2.22 ± 0.36 |
| KungFuMaster | 0.13 ± 0.07 | 0.22 ± 0.05 | 0.01 ± 0.00 | 0.45 ± 0.03 | **0.48 ± 0.04** | 0.37 ± 0.04 | 0.17 ± 0.10 | 0.14 ± 0.11 | 0.41 ± 0.05 |
| MsPacman | 0.11 ± 0.00 | 0.13 ± 0.01 | 0.00 ± 0.00 | 0.17 ± 0.01 | **0.20 ± 0.02** | 0.11 ± 0.02 | 0.08 ± 0.05 | 0.07 ± 0.05 | 0.15 ± 0.01 |
| Pong | 0.33 ± 0.30 | 0.31 ± 0.39 | -0.02 ± 0.00 | **0.99 ± 0.16** | 0.96 ± 0.24 | 0.17 ± 0.09 | 0.16 ± 0.18 | 0.06 ± 0.10 | 0.71 ± 0.15 |
| PrivateEye | 0.00 ± 0.00 | **0.00 ± 0.00** | -0.00 ± 0.00 | 0.00 ± 0.00 | 0.00 ± 0.00 | -0.00 ± 0.00 | -0.00 ± 0.00 | -0.00 ± 0.00 | 0.00 ± 0.00 |
| Qbert | 0.03 ± 0.01 | 0.04 ± 0.01 | -0.00 ± 0.00 | 0.05 ± 0.03 | 0.08 ± 0.04 | **0.09 ± 0.02** | 0.02 ± 0.01 | 0.02 ± 0.01 | 0.04 ± 0.01 |
| RoadRunner | 0.60 ± 0.09 | 0.67 ± 0.02 | 0.11 ± 0.01 | 0.82 ± 0.47 | **1.00 ± 0.22** | 0.49 ± 0.28 | 0.39 ± 0.14 | 0.37 ± 0.12 | 0.60 ± 0.01 |
| Seaquest | 0.02 ± 0.00 | 0.03 ± 0.00 | 0.00 ± 0.00 | 0.03 ± 0.01 | **0.03 ± 0.01** | 0.02 ± 0.01 | 0.01 ± 0.01 | 0.01 ± 0.00 | 0.02 ± 0.00 |
| UpNDown | 0.23 ± 0.01 | 0.39 ± 0.17 | 0.15 ± 0.00 | 1.75 ± 0.15 | **2.17 ± 0.54** | 1.92 ± 0.58 | 0.14 ± 0.03 | 0.14 ± 0.01 | 1.18 ± 0.35 |
| **Aggregate** | 0.29 ± 0.52 | 0.35 ± 0.53 | 0.04 ± 0.28 | 1.13 ± 1.19 | **1.25 ± 1.55** | 0.48 ± 0.64 | 0.28 ± 0.35 | 0.24 ± 0.29 | 0.61 ± 0.67 |

*Table 12.* Atari mean scores after training for 100M frames (Human Normalized)

| Environment | dqn | dqn+spr | qrc | qrc+spr+orth | strq | strq+spr+orth$^2$ |
|---|---|---|---|---|---|---|
| Alien | $0.12 \pm 0.01$ | $0.11 \pm 0.01$ | $-0.01 \pm 0.00$ | $\mathbf{0.15 \pm 0.00}$ | $0.13 \pm 0.00$ | $0.14 \pm 0.01$ |
| Amidar | $0.10 \pm 0.00$ | $0.12 \pm 0.01$ | $0.03 \pm 0.00$ | $\mathbf{0.20 \pm 0.02}$ | $0.15 \pm 0.02$ | $0.16 \pm 0.01$ |
| Assault | $2.34 \pm 0.30$ | $1.49 \pm 1.09$ | $0.21 \pm 0.02$ | $\mathbf{4.45 \pm 0.27}$ | $2.56 \pm 0.26$ | $2.33 \pm 0.16$ |
| Asterix | $0.17 \pm 0.01$ | $0.16 \pm 0.02$ | $0.03 \pm 0.00$ | $\mathbf{0.26 \pm 0.04}$ | $0.16 \pm 0.02$ | $0.18 \pm 0.01$ |
| BankHeist | $-0.01 \pm 0.00$ | $0.09 \pm 0.03$ | $-0.00 \pm 0.01$ | $\mathbf{0.64 \pm 0.38}$ | $0.08 \pm 0.03$ | $0.13 \pm 0.04$ |
| BattleZone | $1.29 \pm 0.17$ | $2.08 \pm 0.30$ | $0.18 \pm 0.03$ | $\mathbf{2.56 \pm 0.10}$ | $0.55 \pm 0.79$ | $1.89 \pm 0.32$ |
| Boxing | $0.50 \pm 0.17$ | $0.50 \pm 0.16$ | $-0.15 \pm 0.01$ | $\mathbf{0.63 \pm 0.06}$ | $0.53 \pm 0.08$ | $0.49 \pm 0.08$ |
| Breakout | $1.18 \pm 0.19$ | $1.88 \pm 0.17$ | $0.06 \pm 0.00$ | $\mathbf{4.51 \pm 0.33}$ | $2.20 \pm 0.18$ | $2.32 \pm 0.26$ |
| ChopperCommand | $0.07 \pm 0.01$ | $0.22 \pm 0.04$ | $0.03 \pm 0.00$ | $0.24 \pm 0.27$ | $0.24 \pm 0.05$ | $\mathbf{0.26 \pm 0.07}$ |
| CrazyClimber | $1.31 \pm 0.05$ | $2.40 \pm 0.11$ | $0.29 \pm 0.01$ | $\mathbf{3.89 \pm 0.30}$ | $3.04 \pm 0.12$ | $3.10 \pm 0.30$ |
| DemonAttack | $0.24 \pm 0.03$ | $\mathbf{0.74 \pm 0.14}$ | $0.08 \pm 0.01$ | $0.55 \pm 0.04$ | $0.16 \pm 0.04$ | $0.27 \pm 0.02$ |
| Freeway | $0.00 \pm 0.00$ | $0.79 \pm 0.40$ | $0.58 \pm 0.29$ | $\mathbf{1.08 \pm 0.01}$ | $0.40 \pm 0.49$ | $0.93 \pm 0.03$ |
| Frostbite | $0.12 \pm 0.09$ | $0.05 \pm 0.01$ | $0.01 \pm 0.01$ | $\mathbf{0.27 \pm 0.11}$ | $0.14 \pm 0.14$ | $0.23 \pm 0.08$ |
| Gopher | $0.35 \pm 0.11$ | $0.43 \pm 0.21$ | $-0.06 \pm 0.01$ | $\mathbf{1.48 \pm 0.33}$ | $1.09 \pm 0.54$ | $0.41 \pm 0.08$ |
| Hero | $0.15 \pm 0.03$ | $0.20 \pm 0.01$ | $-0.03 \pm 0.00$ | $0.40 \pm 0.09$ | $0.33 \pm 0.06$ | $\mathbf{0.45 \pm 0.05}$ |
| Jamesbond | $0.90 \pm 0.40$ | $1.00 \pm 0.87$ | $0.53 \pm 0.20$ | $\mathbf{1.83 \pm 0.47}$ | $1.37 \pm 0.21$ | $1.42 \pm 0.31$ |
| Kangaroo | $3.16 \pm 1.94$ | $3.00 \pm 2.65$ | $0.33 \pm 0.05$ | $\mathbf{8.88 \pm 0.79}$ | $1.67 \pm 1.43$ | $2.38 \pm 2.68$ |
| Krull | $0.72 \pm 0.29$ | $0.89 \pm 0.85$ | $-1.08 \pm 0.06$ | $\mathbf{4.16 \pm 0.56}$ | $3.04 \pm 0.37$ | $2.67 \pm 0.46$ |
| KungFuMaster | $0.23 \pm 0.07$ | $0.34 \pm 0.01$ | $0.01 \pm 0.00$ | $\mathbf{0.54 \pm 0.04}$ | $0.50 \pm 0.03$ | $0.52 \pm 0.03$ |
| MsPacman | $0.13 \pm 0.01$ | $0.16 \pm 0.01$ | $0.00 \pm 0.00$ | $\mathbf{0.23 \pm 0.02}$ | $0.13 \pm 0.02$ | $0.18 \pm 0.01$ |
| Pong | $0.82 \pm 0.31$ | $0.81 \pm 0.29$ | $-0.02 \pm 0.00$ | $0.81 \pm 0.43$ | $0.60 \pm 0.20$ | $\mathbf{0.93 \pm 0.24}$ |
| PrivateEye | $0.00 \pm 0.00$ | $0.00 \pm 0.00$ | $-0.00 \pm 0.00$ | $\mathbf{0.00 \pm 0.00}$ | $-0.00 \pm 0.00$ | $0.00 \pm 0.00$ |
| Qbert | $0.07 \pm 0.01$ | $0.10 \pm 0.01$ | $-0.00 \pm 0.00$ | $\mathbf{0.13 \pm 0.01}$ | $0.12 \pm 0.03$ | $0.07 \pm 0.02$ |
| RoadRunner | $1.09 \pm 0.05$ | $1.02 \pm 0.12$ | $0.10 \pm 0.02$ | $\mathbf{1.10 \pm 0.20}$ | $0.80 \pm 0.17$ | $0.70 \pm 0.11$ |
| Seaquest | $0.03 \pm 0.01$ | $0.03 \pm 0.00$ | $0.00 \pm 0.00$ | $\mathbf{0.05 \pm 0.01}$ | $0.02 \pm 0.01$ | $0.03 \pm 0.00$ |
| UpNDown | $0.31 \pm 0.06$ | $0.85 \pm 0.94$ | $0.15 \pm 0.01$ | $\mathbf{2.83 \pm 0.66}$ | $1.71 \pm 0.32$ | $1.71 \pm 0.60$ |
| **Aggregate** | $0.59 \pm 0.76$ | $0.75 \pm 0.79$ | $0.05 \pm 0.28$ | $\mathbf{1.61 \pm 2.05}$ | $0.79 \pm 0.88$ | $0.91 \pm 0.93$ |

Figure 6 presents the performance profiles on Atari (40M frames). It plots the probability of achieving a certain score versus the score taking into account all experiment runs. We observe that for both QRC($\lambda$) and Streaming DQN, the addition of SPR significantly shifts the curve to the right, indicating a higher probability of achieving better scores across the entire suite.

## C. Gradient Orthogonalization

Gradient Orthogonalization (Han et al., 2025) is an optimization technique designed to address the "catastrophic correlation" problem inherent in streaming data. Standard optimizers like SGD or AdamW (Loshchilov & Hutter, 2019) assume that gradients computed from consecutive mini-batches are independent and identically distributed (IID). However, in streaming settings (e.g., continuous video or agent trajectories), consecutive gradients are highly correlated ($\cos(g_t, g_{t-1}) \approx 1$), where $g_t$ is the gradient at time $t$. This correlation causes standard optimizers to over-step in the dominant direction of the trajectory, leading to overfitting and representational collapse.

To decorrelate the learning signal, the method filters out the component of the current gradient that is redundant with respect to the recent history. It maintains an exponential moving average (EMA) of the "clean" raw gradients, denoted as $c_t$, to track the dominant descent direction:

$$c_t = \beta_{orth} c_{t-1} + (1 - \beta_{orth}) \nabla_\theta \mathcal{L}_t \tag{6}$$

where $\beta_{orth} \in [0, 1)$ is a momentum coefficient (typically 0.9). At each step, the current gradient $g_t = \nabla_\theta \mathcal{L}_t$ is projected onto the subspace orthogonal to the momentum history $c_{t-1}$. The update vector $u_t$ is computed as:

$$u_t = g_t - \text{proj}_{c_{t-1}}(g_t) = g_t - \frac{g_t \cdot c_{t-1}}{\|c_{t-1}\|^2 + \epsilon} c_{t-1} \tag{7}$$

This orthogonalized gradient $u_t$ replaces the raw gradient $g_t$ in the base optimizer update step. Geometrically, this ensures that the model only updates on "novel" information. If the stream is highly correlated, the projection reduces the magnitude

of the update effectively to zero, preventing the optimizer from continuously pushing weights in the same direction. Conversely, if the data is IID, $u_t \approx g_t$, and the algorithm behaves like a standard optimizer.

## D. The QRC($\lambda$) Agent

QRC($\lambda$) (Elelimy et al., 2025) is a value-based Reinforcement Learning algorithm designed to optimize the Projected Bellman Error ($\overline{\text{PBE}}$) (Patterson et al., 2022) in a streaming setting. Standard Q-learning and its deep variants minimize the mean squared Bellman Error ($\overline{\text{BE}}$) using semi-gradient updates, a procedure that does not correspond to the optimization of a true objective and is known to diverge when combined with non-linear function approximation and off-policy sampling (Baird et al., 1995; Tsitsiklis & Van Roy, 1996). The $\overline{\text{PBE}}$ was proposed as a geometrically sound alternative for value estimation, addressing these instabilities by formulating a well-defined objective while circumventing the double-sampling issue inherent in the $\overline{\text{BE}}$. This is achieved by introducing an auxiliary function $h_\psi(s, a)$ parameterized by $\psi$, which estimates the expected TD-error and avoids the need for a second independent sample of the environment transition.

Building on this framework, Elelimy et al. (2025) extended the $\overline{\text{PBE}}$ to the control setting with multi-step returns, resulting in the QRC($\lambda$) algorithm. QRC($\lambda$) minimizes the $\overline{\text{PBE}}$ using a backward-view eligibility trace mechanism, making it particularly well suited for streaming RL where replay buffers and batch updates are impractical. To ensure stability in this regime, the algorithm incorporates a regularization term on the auxiliary network derived from the TDRC (TD with Regularized Corrections) formulation (Ghiassian et al., 2020), yielding a "Gradient TD" style update that remains stable under continual, on-policy data streams.

To enable multi-step credit assignment without storing past transitions, QRC($\lambda$) extends the $\overline{\text{PBE}}$ to $\lambda$-returns and employs a backward-view mechanism based on eligibility traces. These traces accumulate gradient information over time, allowing parameter updates to depend on a history of visited states while maintaining constant memory. Let $Q_w(s, a)$ denote the primary action-value network and $h_\psi(s, a)$ the auxiliary network. The standard one-step TD error is defined as

$$\delta_t = r_{t+1} + \gamma \max_{a'} Q_w(s_{t+1}, a') - Q_w(s_t, a_t). \tag{8}$$

QRC($\lambda$) maintains three distinct eligibility traces: $z^w$ for the Q-value gradients, $z^h$ for the auxiliary value estimates, and $z^\psi$ for the auxiliary gradients. At each step $t$, these traces are updated as

$$
\begin{aligned}
z_t^w &= \gamma \lambda z_{t-1}^w + \nabla_w Q_w(s_t, a_t), \\
z_t^h &= \gamma \lambda z_{t-1}^h + h_\psi(s_t, a_t), \\
z_t^\psi &= \gamma \lambda z_{t-1}^\psi + \nabla_\psi h_\psi(s_t, a_t).
\end{aligned} \tag{9}
$$

The parameters are then updated using the gradient of the PBE($\lambda$) objective:

$$
\begin{aligned}
\Delta w &= \delta_t z_t^w - h_\psi(s_t, a_t) \nabla_w Q_w(s_t, a_t) - z_t^h \nabla_w \delta_t, \\
\Delta \psi &= \delta_t z_t^\psi - h_\psi(s_t, a_t) \nabla_\psi h_\psi(s_t, a_t) - \beta_{qrc} \psi_t,
\end{aligned} \tag{10}
$$

where $\nabla_w \delta_t = \gamma \nabla_w \max_{a'} Q_w(s_{t+1}, a') - \nabla_w Q_w(s_t, a_t)$ and $\beta_{qrc}$ is a regularization coefficient that prevents the auxiliary network parameters from growing unbounded. Finally, following Watkins' Q($\lambda$) (Watkins et al., 1989), all eligibility traces are reset to zero whenever a non-greedy action is taken to mitigate off-policy discrepancies.

## E. The Stream Q($\lambda$) Agent

This section goes over a detailed explanation of the Stream Q($\lambda$) agent from Elsayed et al. (2024). In the streaming setting, the non-stationarity of the RL framework is exacerbated due to the fact that updates are performed on highly correlated gradients. The idea behind the design of the Stream Q($\lambda$) agent is to address the instability factors that are the most affected by the streaming setting, namely the decreased sampling efficiency, the learning instability due to highly correlated gradients and the poor learning dynamics induced by improper scaling of data.

The authors address sampling efficiency in two ways. First, by using eligibility traces rather than regular gradients for their updates as they provide better and faster credit assignment. Second, they use a sparse initialization scheme in order to promote sparse representations, as they have been shown to reduce forgetting and be beneficial for RL (Lan & Mahmood, 2023).

Because gradients between successive updates are highly correlated, the resulting parameter updates may vary a lot in magnitude. The authors propose a new optimizer, named Overshooting-bounded Gradient Descent (ObGD). As mentioned in the background section, the ObGD optimizer performs a one-step approximation of a backtracking line-search algorithm (Armijo, 1966). In other words, ObGD gives a good candidate value for the highest achievable learning rate such that a stability criterion is verified. In the case of Stream Q($\lambda$), the stability criterion considered is the effective step size defined by Kearney (2023) as:

$$\xi = \frac{\delta(s_t) - \delta_+(s_t)}{\delta(s_t)} \tag{11}$$

where $\delta(s_t)$ is the TD-error and $\delta_+(s_t)$ is the TD-error on the same state after having updated the network parameters according to $\delta(s_t)$. An update is considered unstable if $\xi > 1$. Usually, a backtracking line-search algorithm would iteratively reduce the learning rate until it finds a value that verifies the stability criterion. ObGD approximates this process by performing only one iteration, and without computing $\delta_+(s_t)$ as this can be a costly operation. Elsayed et al. (2024) come up with the following upper bound for the stability criterion: $\xi \leq \alpha\kappa\bar{\delta}_t||\mathbf{z}_t||_1$, where $\alpha$ is the step size, $\kappa > 1$ is an hyperparameter of ObGD, acting as a security coefficient. $\bar{\delta}_t = \max(1, |\delta(s_t)|)$ and $\mathbf{z}_t$ is the eligibility trace used in the update. This combined with the stability criterion gives the following condition on the stability of the learning rate: $\alpha \leq (\kappa\bar{\delta}_t||\mathbf{z}_t||_1)^{-1}$. For a desired maximum step-size $\alpha^*$, ObGD is thereby defined by the following update rule:

$$\mathbf{w}_t \leftarrow \mathbf{w}_t + \min\left(\alpha^*, \frac{1}{\kappa\bar{\delta}_t||\mathbf{z}_t||_1}\right) \cdot \delta(s_t)\mathbf{z}_t \tag{12}$$

The adaptive step-size offered by ObGD allows to efficiently deal with highly correlated updates by ensuring that the behavior of the network after update will be close to its behavior before the update. In order to provide more stability to the network, the authors also use layer normalization before every activation function. They use parameter-free LayerNorm as explained in Appendix A.5.

Finally, the authors normalize the rewards and observations of the agent, keeping a running average and standard deviation of all states and rewards encountered so far. This is a known method to improve training stability (Andrychowicz et al., 2021).

## F. Overview of CURL

Contrastive Unsupervised Representations for Reinforcement Learning (CURL) (Laskin et al., 2020) improves pixel-based RL sample efficiency via an auxiliary contrastive task based on *instance discrimination*. It aligns latent representations of differently augmented views of the same observation (positives) while distinguishing them from other batch samples (negatives). A query encoder $f_\theta$ and a momentum-averaged key encoder $f_\xi$ map augmented views to latents $z_q$ and $z_k$, where $\xi$ is updated via $\xi \leftarrow m\xi + (1-m)\theta$ (He et al., 2020). Using a learned bilinear similarity $\text{sim}(z_q, z_k) = z_q^T W z_k$, CURL minimizes the InfoNCE loss (Oord et al., 2018) for a batch of size $K$:

$$\mathcal{L}_{CURL} = -\log \frac{\exp(z_q^T W z_{k_+})}{\exp(z_q^T W z_{k_+}) + \sum_{i=1}^{K-1} \exp(z_q^T W z_{k_i})} \tag{13}$$

This auxiliary objective drives the shared encoder to extract rich semantic features even when extrinsic rewards are sparse. In our implementation, we used a latent dimension of 128, the contrastive loss weight of 1.0, and a momentum coefficient of 0.95 for key encoder $f_\xi$. We used the same set of augmentations as used by SPR (Appendix A) with the same Q network and encoder architecture, along with stability fixes like layerNorm, spareInit, and reward and observation normalization.

## G. Training Curves

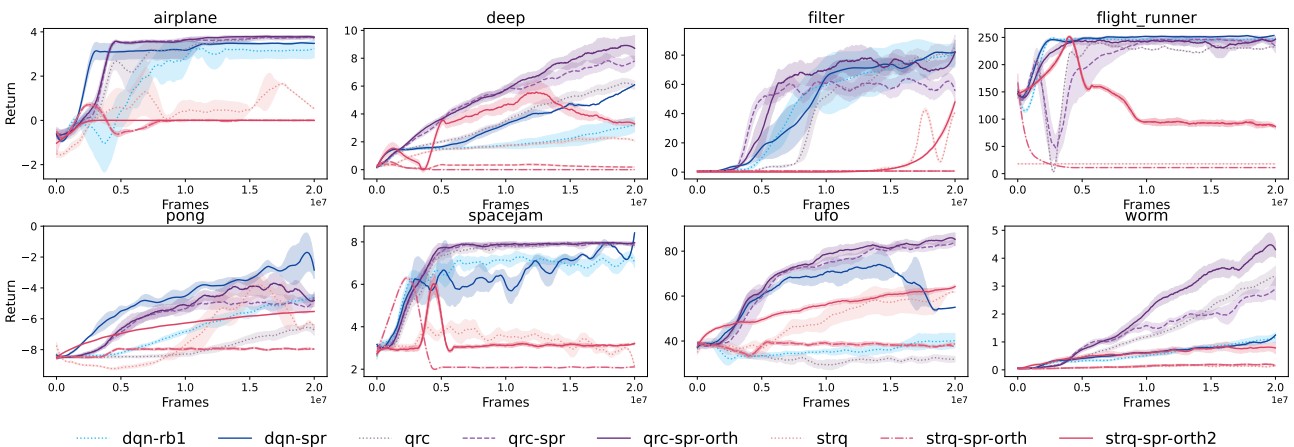

*Figure 7.* Training curves for all octax environments (20M frames)

*Figure 8.* Training curves for all 26 Atari environments (100M frames)

# H. Latent Visualizations

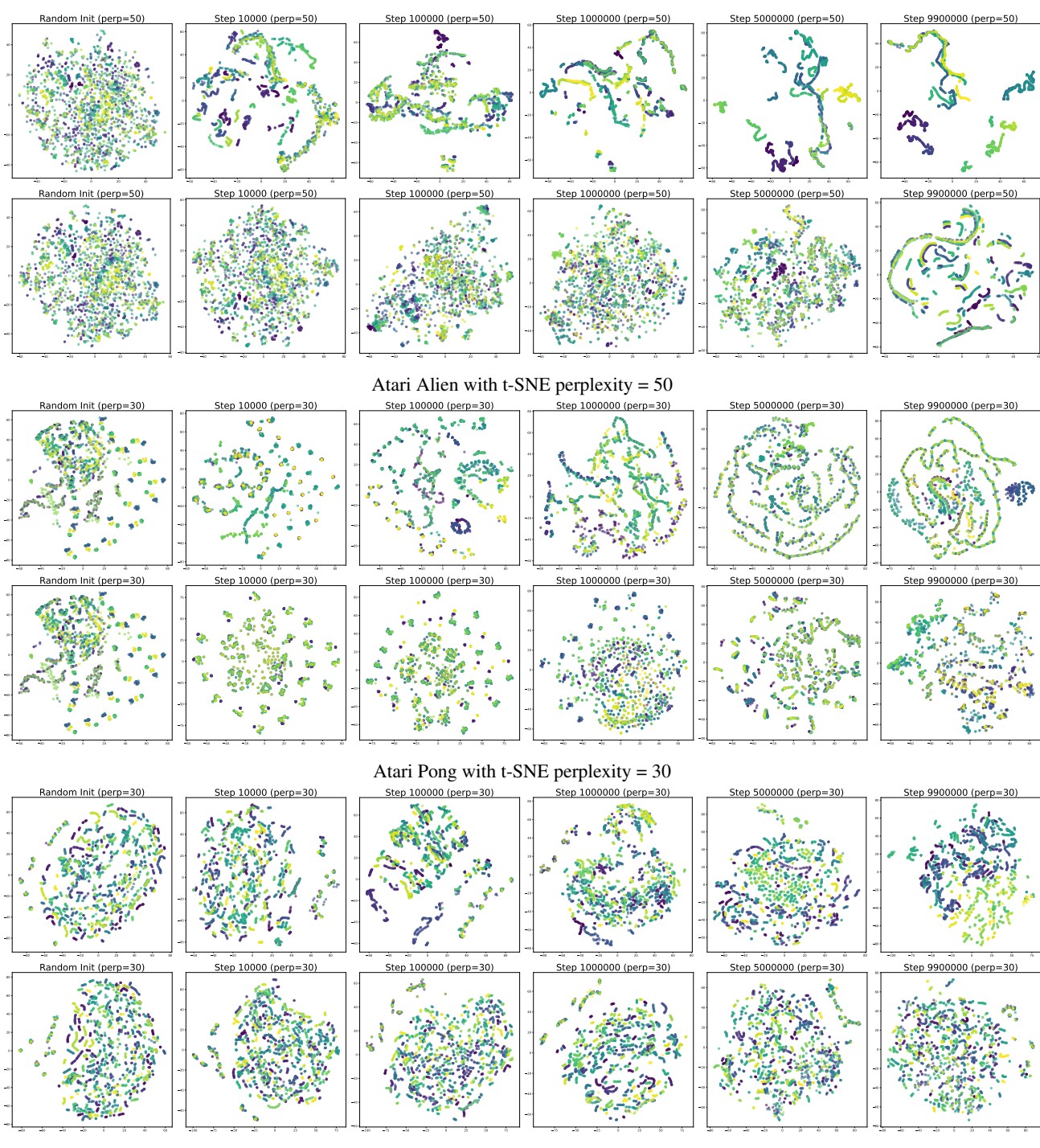

Atari Alien with t-SNE perplexity = 50

Atari Pong with t-SNE perplexity = 30

Atari Assault with t-SNE perplexity = 30

*Figure 9.* We visualize the t-SNE of the encoder latents computed on a random 5K steps trajectory on several environments, with QRC($\lambda$) SPR latents on TOP and QRC($\lambda$) on BOTTOM. They show how latents evolve over the training, with first column being the random initialization and the last column being after 40M frames. Figure 2 showed the same figure on the Alien environment with a t-SNE perplexity of 30, while here the perplexity is set to 50. We also show similar plots for the Pong and the Assault environment. In all cases we see that with SPR, the encoder learns more temporally coherent latents, as shown by the color gradient which indicates time.

