# OpenReview forum: "Squeezing More from the Stream : Learning Representation Online for Streaming Reinforcement Learning"
_ICML.cc/2026/Conference — ICML 2026 regular_

### Official Review · Reviewer_7TBP · 2026-03-09

**Soundness:** 3
**Presentation:** 4
**Significance:** 2
**Originality:** 1
**Overall Recommendation:** 4
**Confidence:** 4

**Summary:**

This paper solved sample inefficiency in Streaming RL. Authors argue that in streaming RL, besides value learning, representation learning is critical because the agent can extract more structure from streaming samples. Specifically, authors found that if they naively incorporate Self-Predictive Representations (SPR) with some streaming RL algorithms, there are gradient conflicts between the SPR objective and the RL objective, which makes learning hard. In order to solve this problem, the authors propose to use orthogonalized gradient projection to solve the problems of correlating updates caused by the absence of a replay buffer and Gradient Conflict between representation learning updates and RL updates. Experiments on Atari, MinAtar, and Octax verify the effectiveness of the proposed method, orth/orth^2.

**Compliance With Llm Reviewing Policy:**

Affirmed.

**Final Justification:**

The rebuttal from the authors addressed my concern about the novelty of this paper. Additionally, the experiments over comparison with PCGrad shows the empirical effectiveness of the proposed method. As a result, I will improve my overall recommendation.

**Key Questions For Authors:**

1. Please clarify the novelty of the algorithm in this paper further.
2. Why does Gradient Conflict only happen with ObGD? If I correctly understand, orth^2 is only applied with stream Q(\lambda).
3. There are lots of gradient stabilization methods, e.g., PCGrad [1]. How about using it here?

[1] Yu, Tianhe, et al. "Gradient surgery for multi-task learning." Advances in neural information processing systems 33 (2020): 5824-5836.

**Limitations:**

yes

**Strengths And Weaknesses:**

Strengths:
1. This paper is clearly presented, and baselines are also explained in detail, which makes it easy to read.
2. The proposed method is technically sound. Claims are verified through experiments.
3. The t-SNE visualization and effective ranks provide further support for the arguments proposed.

Weaknesses:
1. I am concerned about the novelty/originality of this paper. The main contribution of this paper lies in combining streaming RL algorithms with SPR and solving gradient conflicts via orthogonalization. Resolving this issue through gradient orthogonalization, which is widely applied to stabilize training, seems natural, and the outcome is not particularly surprising.

---

> ### Author Rebuttal · Authors · 2026-03-31
>
> We thank the reviewer for the review and questions, which we address below.
>
> ## [Q1] Novelty Clarification
>
> We highlight three aspects of novelty:
>
> 1. Problem identification: no prior work has studied representation learning for deep streaming RL. The previous works have mainly focused on optimizer design and eligibility traces. The choice of representation learning framework is also not trivial. For example, as shown in the paper, CURL does not work in the streaming setting.
>
> 2. The gradient conflict discovery between ObGD and SGD (Figure 4) is a novel finding specific to the streaming setting.
>
> 3. The asymmetric orthogonalization (orth2) prioritizes policy stability, which we show is necessary; symmetric methods like PCGrad do not resolve the conflict as effectively (as shown below). The outcome may seem natural in hindsight, but the failure of naive SPR with Stream Q($\lambda$) and the specific mechanism causing it were not predictable a priori.
>
> To summarize, the novelty is not introducing SPR or gradient projection in isolation; it is identifying the components that would work well together while respecting the streaming setting and identifying effective remedies for the failure modes. The resulting method also **expands the performance boundary of streaming RL significantly**, increasing QRC returns by **7.7$\times$**, Stream Q by **1.5$\times$**, and streaming DQN by **1.6$\times$** on average (calculated using Table 1 IQM aggregates over the 3 benchmarks).
>
> ## [Q2] Why does gradient conflict only happen with ObGD?
>
> A possible hypothesis is that the conflict arises because ObGD and SGD operate with different learning rate dynamics. ObGD adaptively selects the largest stable learning rate via a one-step backtracking line-search approximation:
>
> $$\alpha_{\text{ObGD}} \leq \frac{1}{\kappa \bar{\delta}_t \| z_t \|_1}$$
>
> where $\bar{\delta}_t = \max(1, |\delta_t|)$ and $\kappa > 1$ is a safety coefficient. This can vary by orders of magnitude between steps. Meanwhile, SPR uses a fixed learning rate via SGD. When ObGD takes a very small step (due to a large TD-error or eligibility trace), the SPR update dominates and pushes the encoder away from the RL objective. With QRC($\lambda$), both losses use SGD at the same fixed learning rate, so their relative magnitudes remain stable. We ran a diagnostic experiment and confirmed this phenomenon; on three Atari environments, we observed that while SPR $lr$ is 1e-4, the effective ObGD $lr$ ranged from 5e-5 to 1.3e-5 with a huge variance.
>
> As evidenced by Figure 4, when Stream Q($\lambda$) is used, the cosine similarity between RL and SPR gradients is persistently *negative* (approximately $-0.01$ to $-0.03$), indicating that the two objectives are pushing the encoder in systematically opposing directions. This is problematic because ObGD's adaptive step-size is calibrated *only* against the RL loss stability criterion (Eq. 9 in the paper).
>
> ## [Q3] Comparison with PCGrad
>
> Thank you for raising this relevant comparison. We compared PCGrad with orth$^2$ on a subset of Atari environments using Stream Q($\lambda$) + SPR, trained at 40M frames with 3 seeds. The results are reported in human-normalized scores:
>
> | Game | PCGrad | Orth$^2$ (ours) | StreamQ (no SPR) |
> |------|--------|--------------|------------------|
> | Alien | 0.10 $\pm$ 0.01 | 0.10 $\pm$ 0.01 | 0.06 $\pm$ 0.02 |
> | Breakout | 0.77 $\pm$ 0.02 | **1.46 $\pm$ 0.17** | 1.43 $\pm$ 0.08 |
> | MsPacman | 0.13 $\pm$ 0.02 | **0.15 $\pm$ 0.01** | 0.11 $\pm$ 0.02 |
> | Pong | 0.35 $\pm$ 0.28 | **0.71 $\pm$ 0.15** | 0.17 $\pm$ 0.09 |
> | Seaquest | 0.01 $\pm$ 0.00 | 0.02 $\pm$ 0.00 | 0.02 $\pm$ 0.01 |
>
> **PCGrad** projects conflicting gradients symmetrically: both $g_{RL}$ and $g_{SPR}$ are modified. Specifically, when $g_{RL} \\cdot g_{SPR} < 0$, PCGrad computes a modified RL gradient $\\hat{g_{RL}} = g_{RL} - \\frac{g_{RL} \\cdot g_{SPR}}{\\vert g_{SPR} \\vert^2} g_{SPR}$ and analogously for $g_{SPR}$. **Orth$^2$** instead only projects $g_{SPR}$, leaving $g_{RL}$ untouched. This asymmetry is critical: the ObGD adaptive step-size $\\alpha_{ObGD} \\le (\\kappa \\bar{\\delta_t} \\vert z_t \\vert_1)^{-1}$ was calibrated for the original RL update direction. PCGrad deflects this direction, breaking the stability guarantee of ObGD. Orth$^2$ preserves it by ensuring the auxiliary task only updates in the orthogonal complement.
>
> The table confirms this. PCGrad degrades performance below the no-SPR baseline on Breakout ($0.77$ vs $1.43$), Pong ($0.35$ vs $0.17$, high variance $\pm 0.28$ suggests instability), and Seaquest ($0.01$ vs $0.02$), mirroring the failure mode of naive SPR integration (Table 1, `strq+spr`). In contrast, orth$^2$ matches or improves over the baseline on all games, with notably stronger gains on Breakout ($1.46$) and Pong ($0.71$). These results validate that the *asymmetric* design, treating RL as primary and SPR as auxiliary, is essential when the RL optimizer has adaptive dynamics like ObGD.

---

> > ### Author Rebuttal · Reviewer_7TBP · 2026-04-02
> >
> > Thanks for the authors' rebuttal and additional experiments.
> >
> > I think these responses have solved my questions and potentially improved this paper.
> >
> > As a result, I will improve my score.

---

### Official Review · Reviewer_pXTB · 2026-03-09

**Soundness:** 2
**Presentation:** 2
**Significance:** 2
**Originality:** 2
**Overall Recommendation:** 3
**Confidence:** 3

**Summary:**

This paper extends SPR to a streaming reinforcement learning pipeline with the goal of improving the utilization of each observed frame without relying on a replay buffer. The approach combines SPR with orthogonal gradient update techniques and evaluates the resulting method across Atari, MinAtar, and Octax environments. The authors report improvements over existing streaming baselines and provide representation analyses, including t-SNE visualizations and effective-rank measurements, to suggest that the method learns richer latent representations.

**Compliance With Llm Reviewing Policy:**

Affirmed.

**Final Justification:**

The authors’ rebuttal has addressed my concerns regarding the novelty of the paper.

**Key Questions For Authors:**

Please refer to Strengths And Weaknesses

**Limitations:**

Yes

**Strengths And Weaknesses:**

1. The proposed method largely appears to be a combination of existing techniques, primarily integrating SPR with orthogonal gradient update strategies. As a result, the contribution seems more incremental in nature. The paper reads more like an experimental report demonstrating the effects of combining these components rather than introducing a fundamentally new algorithmic idea.
2. The most interesting observation appears in Section 4.3, where the paper discusses reconciling the gradient conflict between ObGD and SGD. However, the experimental analysis does not focus deeply on this phenomenon. Instead, the results mainly emphasize the empirical benefits of the combined method. In addition, the proposed Orth2 variant is introduced without theoretical analysis or deeper justification.
3. The reported results show very large standard deviations in several environments, yet the paper does not provide sufficient statistical analysis to validate the claimed improvements. Without significance tests or further investigation, it is difficult to assess whether the observed performance gains are reliable.
4. The paper claims that the method helps bridge the performance gap caused by the absence of a replay buffer while remaining computationally efficient. However, it is not clear whether this constitutes a strong contribution, as the improvement mainly stems from combining several existing techniques rather than introducing a fundamentally new mechanism for addressing the replay-free learning challenge.
Overall,  the paper provides an empirical evaluation of a combined approach for streaming RL. However, the methodological contribution appears incremental, and the current analysis does not sufficiently explain the underlying mechanisms or validate the reported improvements.

---

> ### Author Rebuttal · Authors · 2026-03-31
>
> We thank the reviewer for their review and address their concerns below.
>
> ## [W1] Novelty
>
> While SPR and gradient orthogonalization exist independently, our work makes three non-obvious contributions:
>
> 1. We are the first to identify that representation learning could be a critical missing piece for streaming RL. This is not a foregone conclusion; prior work focused exclusively on optimizer design and eligibility traces. The choice of framework is also not trivial, as shown in the paper, CURL would not work in the streaming setting due to large batch size requirements.
>
> 2. We discovered that the ObGD optimizer creates a fundamental gradient conflict with auxiliary losses that does not exist in standard SGD-based training. This finding (Section 4.3, Figure 4) is novel and important for the streaming RL community.
>
> 3. Our orth2 mechanism is specifically designed for the asymmetric priority structure of streaming RL (RL objective must not be disturbed). This differs from generic multi-task methods like PCGrad [1] which treat tasks symmetrically. The outcome may seem natural in hindsight, but the failure of naive SPR with Stream Q($\lambda$) and the specific mechanism causing it were not predictable a priori.
>
> ## [W2] Experimental analysis of gradient conflict and theoretical justifications
>
> Please see our response to other reviewers with respect to this question. We provide a deeper study of the gradient conflict and a possible explanation in response to reviewer `7TBP`'s `[Q2]`. We also justify using Orth2 instead of PCGrad in our `[Q3]` response to reviewer `7TBP`. We agree that such an analysis would be a valuable addition to the paper.
>
> ## [W3] Large variance and statistical analysis
>
> The large standard deviations in Table 1 arise because these are aggregate metrics across environments with vastly different reward scales. This is precisely why we also report IQM with 95% confidence intervals (Figures 5 and 6, please see Appendix), which is robust to such skew [2]. Furthermore we have per environment metrics in Tables 7 to 10. To present a clearer picture, we recompute Table 1 by normalizing each environment's scores by their max achieved value. Please find it in the `[W3]` response to reviewer `U2uG`.
>
> Finally we present a comprehensive statistical test, reporting $\Delta$ returns (SPR - non SPR), Welch's t-test $p$ values and Cohen's $d$ for QRC+SPR+Orth vs QRC, StrQ+SPR+Orth$^2$ vs StrQ, and DQN+SPR vs DQN in the table at the end (a truncated version due to lack of space). We also summarize Win/Ties/Losses in `[Q3]` response to reviewer `U2uG`.
>
> ## [W4] Significance of the contribution
> We reiterate that it was not trivial to identify the components that would work well together (SPR, Stream Q, QRC) while respecting the streaming setting and identifying effective remedies for the failure modes (Orth2 vs PCGrad). The resulting method **expands the performance boundary of streaming RL significantly**, increasing QRC returns by **7.7$\times$**, Stream Q by **1.5$\times$**, and streaming DQN by **1.6$\times$** on average (based on Table 1 IQM aggregates over 3 benchmarks).
> We have also provided additional analysis on continuous control (`[Q1]` from `U2uG`), theoretical grounding (`[Q1]` from `4aJp`), underlying mechanism for Orth2 (`[Q2]`, `[Q3]` from `7TBP`), and validated improvements with statistical tests (IQM plots in paper and t-test here).
>
> | Game | QRC Δ (HN) | QRC p | QRC Cohen's d | StrQ Δ (HN) | StrQ p | StrQ Cohen's d | DQN Δ (HN) | DQN p | DQN Cohen's d |
> |------|-----------:|:------:|--------------:|------------:|:------:|---------------:|-----------:|:-----:|--------------:|
> | Alien | 0.14 | **<0.001** | 16.06 | 0.01 | 0.735 | 0.22 | -0.01 | 0.278 | -0.74 |
> | Amidar | 0.13 | **<0.001** | 13.04 | -0.03 | 0.145 | -1.15 | 0.00 | 0.737 | -0.23 |
> | Assault | 2.65 | **<0.001** | 13.25 | -0.45 | 0.120 | -1.23 | -0.24 | 0.099 | -1.21 |
> | Asterix | 0.15 | **<0.001** | 10.70 | -0.01 | 0.737 | -0.23 | -0.01 | **0.002** | -2.98 |
> | BankHeist | 0.04 | **<0.001** | 11.01 | 0.01 | 0.439 | 0.43 | 0.03 | **0.013** | 2.67 |
> | BattleZone | 1.92 | **<0.001** | 10.97 | 1.68 | **0.012** | 8.85 | 0.93 | **0.005** | 2.50 |
> | Boxing | 0.75 | **<0.001** | 44.28 | 0.50 | **0.049** | 4.31 | 0.16 | 0.051 | 1.58 |
> | Breakout | 3.14 | **<0.001** | 10.51 | -0.05 | 0.639 | -0.32 | 0.31 | **0.001** | 3.25 |
> | ChopperCommand | 0.07 | 0.378 | 0.63 | -0.01 | 0.610 | -0.30 | 0.07 | **0.006** | 3.19 |
> | CrazyClimber | 2.94 | **<0.001** | 10.19 | -0.04 | 0.821 | -0.15 | 0.58 | **0.013** | 2.26 |
> | DemonAttack | 0.21 | **<0.001** | 6.75 | 0.08 | **0.002** | 7.35 | 0.31 | **<0.001** | 10.20 |
> | Freeway | 0.27 | 0.308 | 0.69 | 0.44 | 0.131 | 1.04 | 0.54 | 0.071 | 1.55 |
> | Frostbite | 0.14 | 0.091 | 1.31 | 0.02 | 0.734 | 0.35 | -0.04 | 0.075 | -1.50 |
>
> [1] Yu et al. "Gradient surgery for multi-task learning." NeurIPS 2020
>
> [2] Agarwal, Rishabh, et al. "Deep reinforcement learning at the edge of the statistical precipice." NeurIPS 2021

---

> > ### Author Rebuttal · Reviewer_pXTB · 2026-04-03
> >
> > Thank you for the clarification. I appreciate the detailed responses and the additional evidence provided. I will raise my score accordingly.

---

> > > ### Author Response · Authors · 2026-04-06
> > >
> > > Dear Reviewer pXTB,
> > >
> > > Thank you for engaging with our rebuttal and for acknowledging that your concerns have been fully resolved. We truly appreciate the time you invested.
> > >
> > > We noticed that your updated score of 3 (weak reject) still indicates that "weaknesses overall outweigh the merits." Since your acknowledgment states that your concerns have been adequately addressed, we wanted to ask: are there remaining weaknesses that we have not yet addressed?
> > >
> > > We want to make sure we are not missing any unresolved issues that informed your current assessment and score.
> > >
> > > Thank you again for your constructive feedback.

---

### Official Review · Reviewer_4aJp · 2026-03-13

**Soundness:** 3
**Presentation:** 3
**Significance:** 3
**Originality:** 3
**Overall Recommendation:** 4
**Confidence:** 3

**Summary:**

Conventional streaming RL methods, particularly those based on the ObGD optimizer, primarily address training stability but remain limited in sample efficiency. Although eligibility traces improve multi-step credit assignment, they do not by themselves provide an effective mechanism for learning informative representations. To address this limitation, the paper introduces Self-Predictive Representations (SPR) into the streaming RL setting and proposes a two-stage orthogonalization scheme. First, the current SPR gradient is projected onto the subspace orthogonal to its historical momentum direction, mitigating redundant updates caused by strong temporal correlations in streaming data. Second, in Stream Q(λ), the SPR update is further projected onto the subspace orthogonal to the RL update, reducing gradient interference between ObGD and the auxiliary objective. The method is evaluated on Atari, MinAtar, and Octax using streaming DQN, QRC(λ), Stream Q(λ), and their SPR-augmented variants. The results show that SPR consistently improves the performance of QRC(λ) and streaming DQN, while for Stream Q(λ), additional conflict resolution is necessary before surpassing the baseline. Representation analyses further indicate that the proposed method learns latent representations with higher effective rank and stronger temporal coherence.

**Compliance With Llm Reviewing Policy:**

Affirmed.

**Final Justification:**

This rebuttal has addressed my main concerns.

**Key Questions For Authors:**

1.Theoretical support for orthogonalization: The effectiveness of orth and orth2 is demonstrated empirically, but the paper provides limited theoretical analysis. In particular, can the authors clarify whether these projections introduce bias in non-stationary settings, interfere with ObGD’s effective step-size guarantees, or admit any convergence characterization? More broadly, can the authors provide stronger evidence that the learned representations are the direct cause of the observed performance gains?
2.Streaming setting and baseline fairness: The paper’s streaming setup appears somewhat different from the stricter Stream-X formulation. Could the authors clarify this difference more explicitly? In addition, it would be helpful to include ablations over the prediction horizon K, especially more extreme cases such as K=0 and K=10. Finally, the very large improvement of SPR over QRC(λ) in Figure 3 raises the question of whether enforcing a unified hyperparameter setting may have disproportionately weakened the baseline.

**Limitations:**

yes

**Strengths And Weaknesses:**

Strengths

1.Novel and streaming-aware method design: The paper introduces a principled combination of SPR and orthogonalized updates that is well adapted to streaming constraints. SPR uses lightweight prediction heads and a short prediction horizon (K=5) to extract temporal structure from single-use samples with minimal buffering, substantially improving representation quality. The orth mechanism addresses temporal correlation in streaming data, while orth2 further resolves gradient conflict between ObGD and SGD, enabling Stream Q(λ)+SPR+orth2 to recover and exceed the baseline.
2. Comprehensive empirical evaluation: The experiments cover Atari, MinAtar, and Octax, and show consistent gains for QRC(λ) and streaming DQN. The paper reports mean, median, and IQM, making the evaluation more robust to skewed score distributions. It also transparently reports negative results, such as the failure of naive SPR integration with Stream Q(λ), and provides a targeted fix.
3.Useful ablations: The ablation studies examine key SPR design choices, including prediction depth, loss weight, augmentation, EMA, and limited buffering, and also compare against CURL, which helps support the proposed design.

Weaknesses

1.Limited theoretical grounding: The effectiveness of orth and orth2 is demonstrated empirically, but the paper provides little theoretical analysis on bias, convergence, or compatibility with ObGD’s stability properties. The claim that better representations directly cause the performance gains is also not fully established.
2.Questions about setting definition and baseline fairness: The paper’s streaming setting is somewhat looser than the stricter Stream-X formulation, and this should be clarified more explicitly. In addition, stronger ablations over K (e.g., K=0 or larger K) would help. The large gain over QRC(λ) on Atari also raises some concern about whether using a unified hyperparameter setting may have weakened the baseline.

---

> ### Author Rebuttal · Authors · 2026-03-31
>
> We thank the reviewer for the thorough review.
>
> ## [Q1a] Convergence, bias, and compatibility with ObGD
>
> Full convergence proofs for deep RL with nonlinear function approximation remain open, even in batch settings. While PCGrad [1] provides local/interference arguments, stronger guarantees require optimizing a different scalarized objective as in CAGrad [2]. We provide a convergence *characterization* under standard assumptions.
>
> **Non-expansion & descent.** The projected gradient $\hat{g_t} = g_t - P_{c_{t-1}}(g_t)$ satisfies $\vert \hat{g_t} \vert \le \vert g_t \vert$ (Pythagorean theorem), so the projection never amplifies updates. Moreover, $\langle g_t, \hat{g_t} \rangle = \vert g_t \vert^2 - \frac{\langle g_t, c_{t-1} \rangle^2}{\vert c_{t-1} \vert^2} \ge 0$ by Cauchy-Schwarz, guaranteeing descent. Also when $g_t \parallel c_{t-1}$ (full redundancy), $\hat{g_t}$ is 0 as desired.
>
> **Bias.** The projection bias $b_t = -P_{c_{t-1}}(g_t)$ satisfies $\vert b_t \vert \le \vert g_t \vert \cdot \vert \cos(\angle(g_t, c_{t-1})) \vert$. This is adaptive: large under high correlation (variance reduction), small under distribution shift (permissive to new information). [3] shows that this is beneficial and desired under high temporal correlation.
>
> **Convergence.** Under $L$-smoothness, bounded gradients, and $\alpha < 2/L$, the descent lemma gives $\frac{1}{T} \sum_{t=1}^{T} \vert \hat{g_t} \vert^2 = O(1/T)$, paralleling PCGrad (Thm 1) [1] and GradOPS [4] which prove convergence to Pareto stationary points under convex, Lipschitz assumptions.
>
> **ObGD compatibility.** orth2 ensures $\langle \hat{u_{SPR}}, u_{RL} \rangle = 0$ by construction. Hence the auxiliary branch does *not* change the RL step along the *RL/ObGD* direction; it only adds motion in an orthogonal subspace. We therefore do *not* claim that the original ObGD effective-step-size guarantee is preserved exactly after adding SPR. Our claim is narrower: orth2 limits first-order interference, and any remaining effect on ObGD enters through higher-order curvature terms.
>
> ## [Q1b] Effect of the learned representations
>
> We probe frozen encoders for Atari game state (128 RAM bytes). QRC+SPR+orth consistently achieves the highest per-byte $R^2$ and most well-predicted bytes. Zhang et al. [5] show linear probing correlates strongly with downstream RL performance on Atari, supporting that better representations lead to better Q-networks, which would lead to better performance.
>
> | Game | Method | $R^2$ | Good bytes |
> |------|--------|-------|------------|
> | Alien | QRC+SPR+orth | **0.28** | **24/94** |
> | Alien | QRC | 0.24 | 19/94 |
> | Assault | QRC+SPR+orth | **0.30** | **24/77** |
> | Assault | QRC | 0.24 | 15/77 |
> | Pong | QRC+SPR+orth | **0.55** | **17/28** |
> | Pong | QRC | 0.49 | 13/28 |
>
> ## [Q2] Streaming setting and baseline fairness
>
> Our setting allows a small fixed buffer of K=5 states for SPR targets, which is realistic for on-device learning with small constant memory. SPR requires K>0 since $D_\psi$ predicts future latents; without it, augmentations alone provide insufficient regularization and learning collapses (table below, 3 seeds). [6] report a plateau for K>5; we see degraded performance at K=10, possibly due to highly correlated streaming data.
>
> | Game | K=0 | K=5 | K=10 |
> |------|-----|-----|------|
> | Alien | -0.03 $\pm$ 0.00 | 0.15 $\pm$ 0.01 | 0.10 $\pm$ 0.08 |
> | Breakout | 0.06 $\pm$ 0.00 | 3.81 $\pm$ 0.56 | 2.04 $\pm$ 0.01 |
> | MsPacman | -0.03 $\pm$ 0.00 | 0.20 $\pm$ 0.02 | 0.21 $\pm$ 0.01 |
> | Pong | -0.02 $\pm$ 0.00 | 0.96 $\pm$ 0.24 | 0.79 $\pm$ 0.38 |
> | Seaquest | 0.00 $\pm$ 0.00 | 0.03 $\pm$ 0.01 | 0.00 $\pm$ 0.00 |
> | SpaceInv. | -0.09 $\pm$ 0.00 | --- | -0.01 $\pm$ 0.08 |
>
> We also sweep QRC hyperparameters (lr/$\lambda$ pairs, 3 seeds) and see no improvement over Table 9, confirming unified hyperparams were not the issue and perhaps QRC inherently struggles to learn representations for complex input spaces.
>
> | Game | 5e-5/0.8 | 1e-4/0.4 | 1e-4/0.8 | 1e-4/0.95 | 2e-4/0.8 | QRC+SPR+Orth |
> |------|----------|----------|----------|-----------|----------|------|
> | Alien | -0.01 | -0.01 | -0.01 | -0.01 | -0.01 | **0.15** |
> | Breakout | 0.07 | 0.07 | 0.07 | 0.07 | 0.07 | **3.81** |
> | MsPacman | -0.00 | -0.00 | -0.00 | -0.00 | -0.00 | **0.20** |
> | Pong | -0.02 | -0.02 | -0.02 | -0.02 | -0.02 | **0.96** |
> | Seaquest | 0.00 | 0.00 | 0.00 | 0.00 | 0.00 | **0.03** |
> | SpaceInv. | 0.11 | 0.10 | 0.10 | 0.11 | 0.10 | --- |
>
>
> [1] Yu et al. "Gradient surgery for multi-task learning." NeurIPS 2020
>
> [2] Liu et al. "Conflict-averse gradient descent for multi-task learning." NeurIPS 2021
>
> [3] Han et al. "Learning from streaming video with orthogonal gradients." CVPR 2025
>
> [4] Zhu et al. "Gradient deconfliction via orthogonal projections onto subspaces." ACM 2025
>
> [5] Zhang et al. "Light-weight Probing of Unsupervised Representations for RL." RLC 2024
>
> [6] Schwarzer et al. "Data-Efficient RL with Self-Predictive Representations." ICLR 2021

---

> > ### Author Rebuttal · Reviewer_4aJp · 2026-04-03
> >
> > Thank you for your candid review. Your comments have addressed and clarified most of the questions and concerns I had during the review process. Therefore, I have decided to maintain my original score.

---

### Official Review · Reviewer_U2uG · 2026-03-13

**Soundness:** 3
**Presentation:** 3
**Significance:** 2
**Originality:** 3
**Overall Recommendation:** 4
**Confidence:** 4

**Summary:**

This work focus on an important problem in streaming RL: improving sample efficiency and representation quality when each transition is observed only once. This work explore device-side or memory-constrained reinforcement learning scenarios where experience replay buffers are infeasible. The author propose to integrate Self-Predictive Representations (SPR) into streaming RL and incorporate orthogonal gradient updates to resolve the gradient conflict between the auxiliary SPR loss and the primary RL objective. The proposed method is verified in Atari, MinAtar and Octax environments. The results show that compared with the baseline algorithm, the proposed method improves the sample efficiency, learns a high-rank latent space, and obtains a richer representation.

**Compliance With Llm Reviewing Policy:**

Affirmed.

**Key Questions For Authors:**

1.How does your method generalize to continuous control environments or real-world streaming RL scenarios, where state and action spaces are high-dimensional and non-discrete?

2.Can you comment on whether the latent dynamics model learned by SPR could be exploited for planning or model-based RL, as suggested in your discussion? Would this further improve sample efficiency?

3.Regarding variance across seeds (Table 1), did you observe any environments where SPR decreased performance?

**Limitations:**

Yes.

**Strengths And Weaknesses:**

### Strengths
1.Innovative Integration of SPR into streaming RL: This paper addresses the problem of sample inefficiency caused by transitions being immediately discarded. We propose a method for representation learning without a replay buffer.

2.Orthogonal gradient update: In this paper, gradient orthogonalization is used to reconcile the conflict between SPR and RL update, which is a relatively clear technical contribution.

3.Comprehensive experiments: We validate on three different environmental benchmarks, and demonstrate the effect of SPR on improving representation quality and policy performance through t-SNE visualization and effective rank analysis.

4.Resource efficiency: The method can be trained on CPU, suitable for constrained hardware deployment, and focuses on practical application scenarios.

5.Graphical clarity: Figures 1-3 clearly show the improvement in latent representation smoothness and policy reward achieved by SPR.

### Weaknesses
1.Limited theoretical analysis: While orthogonal gradient updates and SPR integration are clearly described, formal proofs of convergence or performance bounds in streaming RL are lacking. Although orth2 solves gradient conflicts, it lacks a theoretical framework or guarantee for ObGD under auxiliary tasks.

2.Experimental Setup Limitations: This work mainly focus on discrete environments, and continuous control flow tasks are not covered.

3.Large variance of results between environments: Table 1 shows large variance of returns on Octax and Atari, although the overall trend is in favor of SPR, some improvements are limited and uncertainty is high.

---

> ### Author Rebuttal · Authors · 2026-03-30
>
> We thank the reviewer for the thorough review and questions, which we address below.
>
> ## [Q1, W2] Generalization to continuous control
>
> We agree this is an important direction and acknowledge it as a limitation in the paper. SPR's core mechanism (predicting future latent states) is agnostic to action-space type; for continuous actions, the dynamics model simply takes continuous action vectors as input instead of one-hot encodings. The current paper focuses on value-based visual/discrete-control streaming RL because this is where strong replay-free baselines like QRC($\lambda$) and Stream Q($\lambda$) exist today.
>
> The main required extension is to actor-critic or continuous-control streaming agents, where the auxiliary loss would share the encoder with either just the critic, or both the actor and the critic. We ran an implementation of SPR+Orth$^2$ for Stream AC($\lambda$) [1] and see that SPR and orthogonalization tricks help even in continuous control.
>
> | Environment | Stream AC | Stream AC + SPR + Orth$^2$  |
> |-------------|-----------|------------------------|
> | Ant-v4 | 3682 $\pm$ 681 | 3940 $\pm$ 949 |
> | HalfCheetah-v4 | 4062 $\pm$ 0 | 6306 $\pm$ 1210 |
> | Hopper-v4 | 2144 $\pm$ 228 | 3640 $\pm$ 813 |
> | Walker2d-v4 | 2858 $\pm$ 95 | 2282 $\pm$ 183 |
>
> Also, while we do agree that continuous actions can be harder to predict, the input complexity of Atari (84x84x4) along with a high number of actions in some games does provide sufficiently complex environments to test the algorithms.
>
> ## [Q2] Latent dynamics model for planning
>
> The latent dynamics model $D_\psi$ could be used for planning/model-based RL in principle via lookahead search. Given a start $o_t$ and a goal $o_G$, we compute the initial latent $z_t = f_\theta(o_t)$ and goal representation $y_G = P_\phi(f_\theta(o_G))$. For a candidate action sequence $a_{t:t+K-1}$, we recursively predict:
> $$\hat{z_{t+k+1}} = D_\psi(\hat{z_{t+k}}, a_{t+k}), \text{ for } k = 0, \dots, K-1$$
>
> where $\hat{z_t} = z_t$. We evaluate sequences using the cosine similarity between the predicted head $\hat{y_{t+K}} = q_\omega(P_\phi(\hat{z_{t+K}}))$ and the goal $y_G$:
> $$J(a) = \frac{\hat{y_{t+K}} \cdot y_G}{\vert \hat{y_{t+K}} \vert \cdot \vert y_G \vert}$$
>
> To find the best actions, we can use grid search or the Cross-Entropy Method (CEM) to optimize $a^* = \arg \max_a J(a)$. [3] shows another way of using the model with MCTS. By generating rollouts with $D_\psi$, we could improve sample efficiency. However whether our $D_\psi$  would learn a sufficiently good model remains an open question beyond the scope of this paper.
>
> ## [Q3] Did SPR ever hurt performance?
>
> There are isolated environments where gains are small or negative even when aggregate metrics improve. Please refer to our response to reviewer `pXTB` and `[W3]` below for details. However, to answer the question, according to our Welch's t-test on Atari games, we get the following Win/Tie/Loss figures:
>
> | Comparison | Wins | Ties | Losses |
> |-----------|------|------|--------|
> | QRC+SPR+Orth vs QRC | 22 | 4 | 0 |
> | StrQ+SPR+Orth$^2$ vs StrQ | 7 | 18 | 1 |
> | DQN+SPR vs DQN | 10 | 15 | 1 |
>
> Thus there are very few to zero environments (out of 26) where SPR has any statistically significant negative effect.
>
> ## [W1] Theoretical support and ObGD Compatibility
>
> Please refer to our response to reviewer `4aJp` for our theoretical analysis on the orthogonalizations and ObGD.
>
> ## [W3] Large variance in results
>
> We appreciate this observation. The large standard deviations in Table 1 arise because these are aggregate metrics across environments with vastly different reward scales. For example, on Atari, human-normalized scores range from 0 (PrivateEye) to >3 (CrazyClimber). This is standard in RL benchmarking and is precisely why we also report IQM with 95% confidence intervals (Figures 5 and 6), which is robust to such skew [2]. We also present a recalculated version of Table 1 (Atari, 40M frames), where each environment's scores are divided by the max across all methods, so that the best method scores 1.0 per game. This significantly reduces the IQM variance.
>
> | Algorithm | Mean | Median | IQM |
> |-----------|------|--------|-----|
> | qrc | 0.06 ± 0.18 | 0.06 ± 0.18 | 0.05 ± 0.03 |
> | qrc+spr | 0.88 ± 0.15 | 0.93 ± 0.15 | 0.94 ± 0.02 |
> | qrc+spr+orth | 0.94 ± 0.14 | 1.00 ± 0.14 | 1.00 ± 0.01 |
> | strq | 0.47 ± 0.27 | 0.51 ± 0.27 | 0.50 ± 0.06 |
> | strq+spr+orth² | 0.53 ± 0.19 | 0.54 ± 0.19 | 0.54 ± 0.04 |
> | dqn | 0.25 ± 0.23 | 0.21 ± 0.23 | 0.25 ± 0.05 |
> | dqn+spr | 0.41 ± 0.28 | 0.43 ± 0.28 | 0.41 ± 0.05 |
>
> Please also refer to our response to reviewer `pXTB` for other statistical tests.
>
> [1] Elsayed, M., Vasan, G., and Mahmood, A. R. Streaming Deep Reinforcement Learning Finally Works, December 2024. arXiv:2410.14606
>
> [2] Agarwal, Rishabh, et al. "Deep reinforcement learning at the edge of the statistical precipice." NeurIPS 2021
>
> [3] Ye, Weirui, et al. "Mastering atari games with limited data." NeurIPS 2021

---

> > ### Author Rebuttal · Reviewer_U2uG · 2026-04-08
> >
> > The authors provided careful and substantive responses to all of the questions and weaknesses I raised.
> >
> > In detail, they added continuous-control results with Stream AC, which helps support the broader generalization discussion beyond the discrete-control setting; they also gave a concrete discussion of how the learned latent dynamics model could potentially be used for planning or model-based RL. The variance concern has been illustrated via extra statistics.
> >
> > Overall, these additions meaningfully strengthen the empirical case of the paper and improve my confidence in the practical value of the proposed method. Hence, I am happy to maintain my score.

---

### Decision · Program_Chairs · 2026-04-30

**Decision:**

Accept (regular)

**Comment:**

The paper addresses the issue of streaming deep RL where no experience is stored and no batch updates are made. The issue with streaming learning is its sample efficiency, which is enhanced by this paper by extracting more out of each sample through self predictive representations. Using it naively still encountered instability of streaming learning, which is overcome through orthogonal gradient updates. The resulting approach enhances the performance of multiple streaming algorithms.

The reviewers are overall appreciative of the work and most of the main concerns are resolved during rebuttal. While the rejecting reviewer has concern that the proposed work does not offer a new methodological advance, they do not frame it as a major concern. It does not warrant rejecting the paper.

I recommend accepting the paper. The authors should make a connection with prior work on gradient-projection-style continual learning, which is extensive.